# PROJECT AND FORGET: SOLVING LARGE-SCALE METRIC CONSTRAINED PROBLEMS

## ABSTRACT

Given a set of distances amongst points, determining what metric representation is most "consistent" with the input distances or the metric that best captures the relevant geometric features of the data is a key step in many machine learning algorithms. In this paper, we focus on metric constrained problems, a class of optimization problems with *metric constraints*. In particular, we identify three types of metric constrained problems: metric nearness (Brickell et al. (2008)), weighted correlation clustering on general graphs (Bansal et al. (2004)), and metric learning (Bellet et al. (2013); Davis et al. (2007)). Because of the large number of constraints in these problems, however, these and other researchers have been forced to restrict either the kinds of metrics learned or the size of the problem that can be solved.

We provide an algorithm, PROJECT AND FORGET, that uses Bregman projections with cutting planes, to solve metric constrained problems with many (possibly exponentially) inequality constraints. We also prove that our algorithm converges to the global optimal solution. Additionally, we show that the optimality error ($L_2$ distance of the current iterate to the optimal) asymptotically decays at an exponential rate. We show that using our method we can solve large problem instances of three types of metric constrained problems, out-performing all state of the art methods with respect to CPU times and problem sizes.

## 1 INTRODUCTION

Given a set of distances amongst data points, many machine learning algorithms are considerably "easier" once these distances adhere to a metric. Furthermore, learning what metric is most "consistent" with the input distances or the metric that best captures the relevant geometric features of the data (e.g., the correlation structure in the data) is a key step in efficient, approximation algorithms for classification, clustering, regression, feature selection, etc. Indyk (1999) provides a list of other computational problems such as nearest neighbor search, (approximate) proximity problems, facility location, and a variety of graph problems for which we have efficient approximation algorithms in a general metric space. Given the importance of metric representations of data sets, we focus on metric constrained problems, a class of optimization problems with *metric constraints*; i.e., optimization of a convex function subject to metric constraints, such as the triangle inequality, on all the output variables.

In particular, we identify three types of metric constrained problems: *metric nearness* (Brickell et al. (2008)), *weighted correlation clustering on general graphs* (Bansal et al. (2004)), and *metric learning* (Bellet et al. (2013); Davis et al. (2007)). Briefly, the metric nearness problem seeks the closest metric to a given set of distances, the goal of correlation clustering is to partition nodes in a graph according to their similarity, and metric learning finds a metric on a dataset that is consistent with (dis)similarity information about the data points.

All of these problems can be modeled as constrained convex optimization problems with a large number of constraints. Unfortunately, because of the large number of constraints, using standard optimization techniques, researchers have been forced to restrict either the kinds of metrics learned or the size of the problem that can be solved.

Many of the existing methods for metric constrained problems suffer from some sort of significant drawback that hampers performance or restricts the instance size. Gradient based algorithms such as

projected gradient descent (e.g., Beck & Teboulle (2009); Nesterov (1983)) or Riemannian gradient descent require a projection onto the space of all metrics, which in general, is an intractable problem. One modification of this approach is to subsample the constraints and then project onto the sampled set (see Nedić (2011); Polyak (2001); Wang & Bertsekas (2013); Wang et al. (2015)). For metric constrained problems, however, we have many more constraints than data points, so the condition numbers of the problems are quite high and these algorithms tend to require a large number of iterations.

Another standard approach is to consider the Lagrangian and maintain a KKT type optimality condition. These methods run into two different kinds of problems. First, computing the gradient becomes an intractable problem for methods that maintain the KKT condition by using Newton's method. Examples of such methods include the interior point method and the barrier method. One fix could be to subsample the constraints and only compute those gradients, but this approach runs into the same drawbacks as before. The other option is to incrementally update the Lagrangian, looking at one constraint at a time. These methods, such as Bauschke & Lewis (2000); Iusem (1991); Iusem & De Pierro (1990), traditionally require us to cycle through all the constraints which is not feasible with metric constraints.

One final approach is to use cutting planes. The performance of this method is very heavily dependent on the cut selection process (see Dey & Molinaro (2018); Poirrier & Yu (2019) for deep discussions about the cut selection process). The discovery of Gomory cuts and other subsequent work such as branch and bound, has led to the viability of the cutting plane method for solving mixed integer linear programs. This success has, however, not transferred to other problems. In general, if the cuts are not selected appropriately, the algorithm could potentially take an exponential number of iterations; i.e., add an exponential number of constraints. Thus, to use this method, we must show for each problem that the specific cutting plane selection method results in a feasible algorithm (see Chandrasekaran et al. (2012) for an example).

In this paper, we provide an algorithm, PROJECT AND FORGET, that uses Bregman projections with cutting planes, to solve metric constrained problems with many (possibly exponentially) inequality constraints. In fact, the algorithm is a general purpose one that can solve large constrained convex optimization problems, not only those arising from metric constraints. We also develop a stochastic version of our algorithm. This version is a similar adaptation of the Bregman method, as Nedić (2011); Polyak (2001); Wang & Bertsekas (2013); Wang et al. (2015) are adaptations of the projected gradient method. This version of our algorithm can be used to solve problems where each data point (or pair, triple of data points) form a constraint. The major contributions of our paper is as follows:

1. Using a specific instantiation of the PROJECT AND FORGET algorithm, we solve the weighted correlation clustering problem on a graph with over $130,000$ nodes. To solve this problem with previous methods, we would need to solve a linear program with over $10^{15}$ constraints. Furthermore, we demonstrate our algorithms superiority by outperforming the current state of the art in terms of CPU times.

2. We use our algorithm to develop a new algorithm that solves the metric nearness problem. We show that our algorithm outperforms the current state of the art with respect to CPU time and can be used to solve the problem for non-complete graphs.

3. We use the the stochastic version of our algorithm to develop a new algorithm to solve the information theoretic machine learning problem. We compare this against the standard method and show that in general we require fewer projections to solve the problem. Using our algorithm we can also solve the full version of the convex program presented in Davis et al. (2007) instead of a heuristic approximation. Thus, demonstrating that we can solve larger instances of the problem.

4. Finally, we prove that our algorithm converges to the global optimal solution. Additionally, we show that the optimality error ($L_2$ distance of the current iterate to the optimal) asymptotically decays at an exponential rate. We also show that because of the FORGET step, when the algorithm terminates, the set of constraints that remain remembered are exactly the active constraints. Thus, our algorithm also finds the set of active constraints.

We present the necessary background material and problem formulations in Section 2. In Section 3, we provide a general form of the PROJECT AND FORGET algorithm and detail its theoretical analysis. We instantiate our algorithm to solve three types of metric constrained problems in Section 4 and

highlight the empirical performance. Complete proofs and discussion may be found in the sections in the Appendix.

## 2 PRELIMINARIES

### 2.1 METRIC CONSTRAINED PROBLEMS, GENERAL FORMULATION

**Metric polytope.** To set the stage for our optimization problems, we define the set over which we optimize first. Let $\text{MET}_n \subset \mathbb{R}^{\binom{n}{2}}$ be the space of all metrics on $n$ points. Given a graph $G$ the metric polytope $\text{MET}(G)$ is the projection of $\text{MET}_n$ onto the coordinates given by the edges of $G$ (i.e., we consider distances only between pairs of points that are adjacent in $G$).

It can be easily seen that for any $x \in \mathbb{R}^{\binom{n}{2}}$, we have that $x \in \text{MET}_n(G)$ if and only if $\forall e \in G$, $x(e) \geq 0$ and for every cycle $C$ in $G$ and $\forall e \in C$, we have that $x(e) \leq \sum_{\tilde{e} \in C, \tilde{e} \neq e} x(\tilde{e})$. Therefore, $\text{MET}_n(G)$ can be described as the intersection of exponentially many half-spaces.

**Metric constrained problems.** Now that we have the set over which we want to optimize, we give a general formulation for metric constrained optimization problems: given a strictly convex function $f$, a graph $G$, and a finite family of half-space $\mathcal{H} = \{H_i\}$ such that $H_i = \{x : \langle a_i, x \rangle \leq b_i\}$, we seek the unique point $x^* \in \bigcap_i H_i \cap \text{MET}(G) =: C$ that minimizes $f$. That is, if we set $A$ to be the matrix whose rows are $a_i$ and $b$ be the vector whose coordinates are $b_i$ we seek

$$
\begin{aligned}
\text{minimize} \quad & f(x) \\
\text{subject to} \quad & Ax \leq b \\
& x \in MET(G).
\end{aligned}
\tag{2.1}
$$

The constraints encoded in the matrix $A$ let us impose additional constraints beyond that of a metric. In this paper, $A$ is used only for correlation clustering.

### 2.2 SPECIFIC METRIC CONSTRAINED PROBLEMS

**Metric nearness.** Following Brickell et al. (2008), the metric nearness problem is: given a point $x \in \mathbb{R}^{\binom{n}{2}}$, find the closest (in some $\ell_p$ norm) point $x^* \in \text{MET}_n$ to $x$. This problem is a form of metric learning; see Brickell et al. (2008) for an application to clustering and see Gilbert & Sonthalia (2018) for an application to unsupervised metric learning. I

**Weighted correlation clustering on graphs.** Bansal et al. (2004) introduced correlation clustering. In this problem, we are given a graph $G = (V, E)$ (not necessarily complete) in which each edge $e$ has two non-negative numbers $w^+(e)$ and $w^-(e)$ that indicate the level of similarity and dissimilarity between its nodes. The goal of correlation clustering is to partition the nodes into clusters so as to minimize some objective function. The most common objective is $\sum_{e \in E} w^+(e)x_e + w^-(e)(1 - x_e)$, where $x_e \in \{0, 1\}$ indicates whether the end points of the edge $e$ belong to different cluster. In general, this variant of the problem is NP-hard and many different algorithms have been developed to solve the problem. The best approximation results (with approximation ratios $O(\log n)$ in general, and $O(1)$ for specific cases), however, are obtained by rounding the solution to the following relaxed linear problem

$$
\begin{aligned}
\text{minimize} \quad & \sum_{e \in E} w^+(e)x_e + w^-(e)(1 - x_e) \\
\text{subject to} \quad & x_{ij} \leq x_{ik} + x_{kj} \quad i, j, k = 1, ..., n \\
& x_{ij} \in [0, 1] \quad i, j = 1, ..., n.
\end{aligned}
\tag{2.2}
$$

See Charikar et al. (2005); Emanuel & Fiat (2003) for details. Many special cases, such as when the weights are $\pm 1$ and $G = K_n$, can be solved with faster algorithms (e.g., Ailon et al. (2005)).

**Metric learning.** The final metric constrained problem we consider is metric learning. There are many different versions of this problem (see Bellet et al. (2013); Suárez Díaz et al. (2018) for two different surveys on the topic) but all of the instantiations have a similar formulation: given a data set $X$ and possibly some additional information, learn an appropriate metric on $X$. Many of the existing methods seek a linear map $L$ such that the learned metric is given by

$$
d_C(x, y) = \sqrt{(x - y)^T C(x - y)}
$$

where $C = L^T L$. Our general purpose algorithm can directly learn the appropriate metric without resorting to the above specific form, however, this generalization requires more focus than we can give it in this paper. Instead, we focus on a specific instantiation (that also differs from the above), specifically information theoretic metric learning (ITML), as in Davis et al. (2007). In ITML, we consider $L^T L$ as the covariance matrix of a Gaussian distribution $p(x; C)$ and we are given two sets $S, D$ which represent the set of similar and dissimilar points. The problem we solve is:

$$
\begin{aligned}
\text{minimize} \quad & KL\big(p(x; C) \| p(x; I)\big) \\
\text{subject to} \quad & d_A(x_i, x_j) \leq u \quad (i, j) \in S \\
& d_A(x_i, x_j) \geq l. \quad (i, j) \in D
\end{aligned}
\tag{2.3}
$$

### 2.3 BREGMAN PROJECTIONS

All of the above problems can be couched in general terms and, in the Appendix 6, we give such a general formulation. We seek to optimize a rich class of convex functions $f$, known as Bregman functions, denoted $\mathcal{B}(S)$, with useful properties for algorithmic and convergence analysis, subject to metric constraints. More details about Bregman functions and the general setting in which our algorithm works can be found in the Appendix sec:generalProblem . We do, however, detail Bregman projections, a key step in our specific algorithms in this section.

**Definition 1.** *Given a convex function $f(x) : S \to \mathbb{R}$ whose gradient is defined on all of $S$, we define its generalized Bregman distance $D_f : S \times S \to \mathbb{R}$ as follows: $D_f(x, y) = f(x) - f(y) - \langle \nabla f(y), x - y \rangle$.*

**Definition 2.** *Given a strictly convex function $f$, a closed convex set $C$, and a point $y$, the projection of $y$ onto $C$ with respect to $D_f$ is a point $x^* \in \text{dom}(f)$ such that*

$$
x^* = \underset{x \in C \cap \text{dom}(f)}{\arg \min} \ D_f(x, y).
$$

## 3 ALGORITHMS AND ANALYSIS

In this section, we present our algorithm that we will use to solve metric constrained problems and state its convergence behavior. The detailed proofs can all be found in the Appendix 8.

### 3.1 THE ALGORITHM

Our method is an iterative one and is presented in Algorithm 1. Each iteration consists of three phases. In the first phase, we obtain[1] a list of violated metric constraints $L$. If we have additional constraints represented in $A$, $\mathcal{A}$ is a list corresponding to these hyperplanes. In the second phase, we merge $L^{(\nu)}$, the list of constraints we have been keeping tracking of up to the $\nu$th iteration, with $L$ and project onto each of the constraints in the list $L^{(\nu)} \cup L \cup \mathcal{A}$ iteratively. Finally, in the third phase, we forget some constraints.

The project and forget steps for algorithm are presented in Algorithm 2. Let us step through the code to understand intuitively its behavior. Let $H_i = \{x : \langle a_i, x \rangle \leq b_i\}$ be a constraint and $x$ the current iterate. The first step is to calculate $x^*$ and $\theta$. Here $x^*$ is the projection of $x$ onto the boundary of $H_i$ and $\theta$ is a "measure" of how far $x$ is from $x^*$. However, $\theta$ can be any real number and so we examine two cases: $\theta$ positive or negative.

It can be easily seen that $\theta$ is negative if and only if the constraint is violated. In this case, we have $c = \theta$ because (as we will see in proof) the algorithm always maintains $z_i \geq 0$. Then on line 5, we compute the projection of $x$ onto $H_i$. Finally, since we corrected $x$ for this constraint, we add $|\theta|$ to $z_i$. Since each time we correct for $H_i$, we add to $z_i$, we see that $z_i$ stores the total corrections made for $H_i$.

On the other hand, if $\theta$ is positive, this constraint is satisfied. In this case, if we also have that $z_i$ is positive; i.e., we have corrected for $H_i$ before, then we have over compensated for this constraint and,

---

[1]In the general formulation of the algorithm in the Appendix 6, this list is obtained by querying a separation oracle with one of two properties. See the Appendix for a detailed discussion.

---

**Algorithm 1** General Algorithm.

---

1: **function** F($f, \mathcal{A}$)
2:      $L^{(0)} = \emptyset$, $z^{(0)} = 0$. Initialize $x^{(0)}$ so that $\nabla f(x^{(0)}) = 0$.
3:      **while** Not Converged **do**
4:          $L = $ METRIC VIOLATIONS($x^{\nu}$)
5:          $\tilde{L}^{(\nu+1)} = L^{(\nu)} \cup L \cup \mathcal{A}$
6:          $x^{(\nu+1)} = \text{Project}(x^{(\nu)}, \tilde{L}^{(\nu+1)})$
7:          $L^{(\nu+1)} = \text{Forget}(\tilde{L}^{(\nu+1)})$
         **return** $x$
8:
9: **function** METRIC VIOLATIONS($d$)
10:      $L = \emptyset$
11:      Let $d(i, j)$ be the weight of shortest path between nodes $i$ and $j$ or $\infty$ if none exists.
12:      **for** Edge $e = (i, j) \in E$ **do**
13:          **if** $w(i, j) > d(i, j)$ **then**
14:              Let $P$ be the shortest path between $i$ and $j$
15:              Add $C = P \cup \{(i, j)\}$ to $L$
         **return** $L$

---

thus, we must undo some of the corrections. If $c = z_i$, then we undo all of the corrections and $z_i$ is set to 0. Otherwise, if $c = \theta$ we only undo part of the correction.

The forget step is relatively easy: given a constraint $H_i$, we check if $z_i = 0$. If $z_i = 0$, then this means we have not done any net corrections for this constraint and we can forget about it; i.e., delete it from $L^{(\nu)}$.

---

**Algorithm 2** Project and Forget algorithms.

---

1: **function** PROJECT($x, z, L$)
2:      **for** $H_i = \{y : \langle a_i, y \rangle = b_i\} \in L$ **do**
3:          Find $x^*, \theta$ by solving $\nabla f(x^*) - \nabla f(x) = \theta a_i$ and $x^* \in H_i$
4:          $c_i = \min(z_i, \theta)$
5:          $x \leftarrow$ such that $\nabla f(x^{n+1}) - \nabla f(x) = c_i a_i$
6:          $z_i \leftarrow z_i - c_i$
         **return** $x, z$
7: **function** FORGET($x, z, L$)
8:      **for** $H_i = \{x : \langle a_i, x \rangle = b_i\} \in L$ **do**
9:          **if** $z_i == 0$ **then** Forget $H_i$
         **return** $L$

---

In general, calculating the Bregman projection (line 3) cannot be done exactly. See Dhillon & Tropp (2007) for a general method to perform the calculation on line 3 and for an analytic formula for when $f$ is a quadratic function.

### 3.2 CONVERGENCE ANALYSIS

Now that we have specified the algorithm, we establish a few crucial theoretical properties. The first is that our algorithm is guaranteed to converge to the global optimum. In fact, we also show that asymptotically, our error decreases at an exponential rate. These main theoretical results can be summarized by the following theorem.

**Theorem 1.** *If $f \in \mathcal{B}(S)$, $H_i$ are strongly zone consistent with respect to $f^2$, and $\exists x^0 \in S$ such that $\nabla f(x^0) = 0$, then*

     *1. Then any sequence $x^n$ produced by Algorithm 1 converges to the optimal solution of problem 2.1.*

---

[2]This is a technical condition and is discussed in Appendix 6. In the case of our problems this is true.

2. *If $x^*$ is the optimal solution, $f$ is twice differentiable at $x^*$, and the Hessian $H := Hf(x^*)$ is positive semidefinite, then there exists $\rho \in (0, 1)$ such that*

$$\lim_{\nu \to \infty} \frac{\|x^* - x^{\nu+1}\|_H}{\|x^* - x^\nu\|_H} \leq \rho \tag{3.1}$$

*where $\|y\|_H^2 = y^T H y$.*

*Proof.* The proof of this theorem has been moved to the supplementary material section. □

The proof of Theorem 1 also establishes another important theoretical property.

**Proposition 1.** *If $a_i$ is an inactive constraint, then there exists a $N$, such that for all $n \geq N$, we have that $z_i^n = 0$. That is, after some finite time, we never project onto inactive constraints ever again.*

**Corollary 1.** *Under the assumptions for part (2) of Theorem 1, we have that the sequence $z^n \to z^*$ also converges.*

These properties are important as they permit the following interpretation of our algorithm. The algorithm spends the initial few iterations identifying the active constraints from amongst a large number of constraints. The algorithm then spends the remainder of the iterations finding the optimal solution with respect to these constraints. This ability to find the set of active constraints is of the main advantages of our algorithm.

### 3.3 STOCHASTIC VARIANT

In some problems, we do not optimize over the whole of $\text{MET}(G)$ but a subset, as is the case in ITML. In such problems, we have constraints defined using subsets of the data points and as we may have many data points, we may have considerably more constraints than we want to examine. For this reason, we present a stochastic version of our algorithm that can be used in these cases. Instead of calling METRIC VIOLATION to get a list of metric violated constraints, we randomly sample from our constraints. In this version, at each iteration, we choose a random set of constraint and project onto these constraints and the ones we remember from before, and then forget some constraints. The advantage this has over similar stochastic methods such Wang et al. (2015) is that as we sample constraints, the list $L^{(\nu)}$ keeps track of the important constraints that we have seen so far. In this case, we have the following convergence result.

**Theorem 2.** *If $f \in \mathcal{B}(S)$, $H_i$ are strongly zone consistent with respect to $f$, and $\exists x^0 \in S$ such that $\nabla f(x^0) = 0$, then with probability 1 any sequence $x^n$ produced by the stochastic algorithm converges to the optimal solution of problem 2.1. Furthermore, if $x^*$ is the optimal solution, $f$ is twice differentiable at $x^*$, and the Hessian $H := Hf(x^*)$ is positive semidefinite, then there exists $\rho \in (0, 1)$ such that with probability 1,*

$$\lim_{\nu \to \infty} \frac{\|x^* - x^{\nu+1}\|_H}{\|x^* - x^\nu\|_H} \leq \rho.$$

## 4 EXPERIMENTS

To demonstrate the effectiveness of our method in solving metric constrained problems, we solve large instances of each of the problem. More details[3] about each of the experiments can be found in the Appendix 9.

### 4.1 WEIGHTED CORRELATION CLUSTERING ON GENERAL GRAPHS

#### 4.1.1 DENSE GRAPHS

As the LP formulation for correlation clustering in Equation 2.2 has $O(n^3)$ constraints, solving the LP for large $n$ becomes infeasible quickly, in terms of both memory and time. Veldt et al. (2019)

---

[3]All implementations and experiments can be found at `https://www.dropbox.com/sh/lq5nnhi4je2lh89/AABUUW7k5z3lXTSm8x1hhN1Da?dl=0`.

showed that for instances with $n \approx 4000$, standard solvers such as Gurobi ran out of memory on a 100 GB machine. On the other hand, Veldt et al. (2019) develop a method using which they can feasibly solve the problem for $n \approx 11000$. To solve the problem, they transformation problem 2.2 into problem 4.1 for some an appropriately defined $d, \tilde{w}, W$. For general $\gamma$, the solution to problem 4.1 approximates the optimal solution to 2.2. However, for large enough $\gamma$ it has been shown that the two problems are equivalent.

$$\begin{aligned} \text{minimize} \quad & \tilde{w}^T |x - d| + \frac{1}{\gamma}|x - d|^T W |x - d| \\ \text{subject to} \quad & x \in \text{MET}(K_n) \end{aligned} \tag{4.1}$$

This is the version of the LP that we solve with our algorithm. We solve this for four graphs from the Stanford sparse network repository Leskovec & Krevl (2014). Following Veldt et al. (2019), we use the method from Wang et al. (2013) to convert these graphs into instances of weighted correlation clustering on a complete graph. We compare our method against Ruggles et al. (2019), a parallel version of Veldt et al. (2019), in terms of running time, quality of the solutions, and memory usage.

We see from Table 1 that our algorithm takes less time to get a better approximation ratio, but requires more memory per iteration. Our algorithm requires more memory because the initial few iterations find a large number of constraints. Later, in the forget step, the algorithm forgets these constraints until the number of constraints stabilizes at a reasonable level. Hence, our initial memory usage is much larger than our later memory usage.

Table 1: Comparison with Ruggles et al. (2019). We set $\gamma = 1$ and ran until the maximum violation of a metric constraint was smaller than $0.01$. We used a parallel version of the algorithm METRIC VIOLATION as our oracle. All of the computations were done on a machine with 16 physical cores and 13 GB of RAM per core. More details about the experiment can be found in the supplementary material section.

| Graph | | Time (s) | | Opt Ratio | | Avg. mem. / iter. (GiB) | |
|---|---|---|---|---|---|---|---|
| | n | Ours | Ruggles et al. | Ours | Ruggles et al. | Ours | Ruggles et al. |
| CAGrQc | 4158 | 2098 | 5577 | 1.33 | 1.38 | 4.4 | 1.3 |
| Power | 4941 | 1393 | 6082 | 1.33 | 1.37 | 5.9 | 2 |
| CAHepTh | 8638 | 9660 | 35021 | 1.33 | 1.36 | 24 | 8 |
| CAHepPh | 11204 | 71071 | 135568 | 1.33 | 1.46 | 27.5 | 15 |

To see how the number of constraints found by the oracle evolves, we plot the number of constraints found by the oracle and the number of constraints after the forget step for the CA-HepTh graph. This plot can be seen in Figure 1. Figure 1 also shows us, as expected, *the exponential decay of the maximum violation of a metric constraint.*

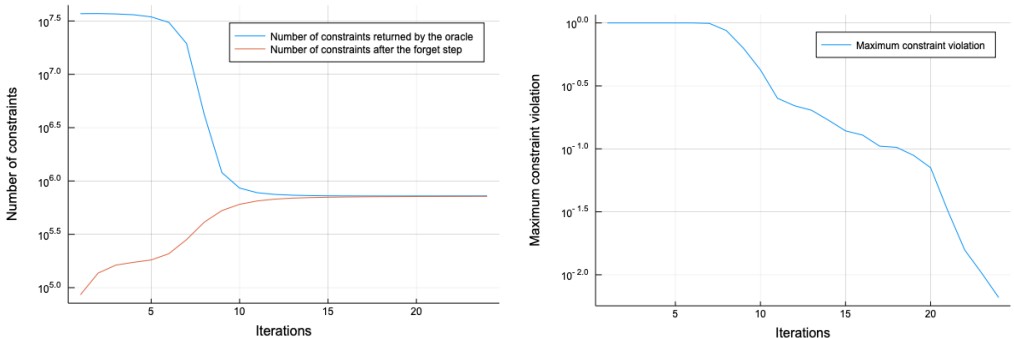

Figure 1: Left: The number of constraints is on the $y$ axis (log scale) and the number of iterations on the $x$ axis. Right: Plot for the maximum violation of a metric constraint (log scale) versus iterations.

### 4.1.2 SPARSE GRAPHS

For much real-world data, the graph $G$ is larger than our previous experiments but it is also sparse. As stated, the problem (2.2) requires $O(n^3)$ constraints and $O(n^2)$ variables. However, in practice, there are very few active constraints, as seen in Table 2. Proposition 2 tells us that if we optimize over $\text{MET}(G)$ instead of $\text{MET}_n$, then the quality of the solution is not degraded. With this generalization to $\text{MET}(G)$, the number of variables is greatly reduced but the number of constraints is increased. Our algorithmic approach, however, can handle this increase in the number of constraints. Thus, we can solve the weighted correlation clustering problem for general graphs $G$.

**Proposition 2.** *Let $\pi$ be the projection from $MET_n$ to $MET(G)$. Then for any optimal solution $x^*$ to the following problem*

$$
\begin{aligned}
&minimize && \sum_{e \in E} w^+(e)x_e + w^-(e)(1 - x_e) \\
&subject\ to && x \in MET(G) \\
& && x_{ij} \in [0, 1], \qquad i, j = 1, ..., n
\end{aligned}
\tag{4.2}
$$

*we have that for all $\hat{x} \in \pi^{-1}(x^*)$, $\hat{x}$ is an optimal solution to 2.2.*

Table 2: Results for our algorithm on larger graphs. We ran our experiment on a machine with 48 physical cores and 13 GB of RAM per core, but we only used 32 threads for the computations.

| Graph | $n$ | # Constraints | Time | Opt Ratio | # Active Constraints | Iters. |
|---|---|---|---|---|---|---|
| Slashdot | 82140 | $5.54 \times 10^{14}$ | 46.7 hours | 1.78 | 384227 | 145 |
| Epinions | 131,828 | $2.29 \times 10^{15}$ | 121.2 hours | 1.77 | 579926 | 193 |

Since the weighting of the edges does not affect the size of the linear program that needs to be solved, we tested our algorithm on signed graphs to get an estimate of the running time for the algorithm. We took two graphs from Leskovec & Krevl (2014). These graphs are much bigger instances than our previous experiments and have 82140 nodes and 131,828 nodes, respectively. Even if we use the parallel version of Veldt et al. (2019), based on the average time it took for a single iteration for the CA-HepPh graph, it would take an estimated two days for a single iteration for a graph with $n \approx 80,000$. Since most graphs require at least 100 iterations, Veldt et al. (2019); Ruggles et al. (2019) cannot be used to solve problems of this magnitude. Other methods of solving the LP are also not feasible as they run out of memory on much smaller instances.

As we can see from Table 2 these instances have over 500 trillion constraints, but the number of active constraints is only a tiny fraction of the number of active constraints. Thus, using our approach we can solve the weighted correlation clustering problem on much larger graphs than ever before and this is possible because the graphs are sparse. That is, our oracle finds violated cycle inequalities relatively quickly and, since we forget inactive constraints, we project onto only a relatively small number of constraints. Thus, each iteration is done relatively quickly. In this experiment, each iteration took 500 to 3000 seconds.

### 4.2 METRIC NEARNESS

To do a head to head comparison against the algorithm presented in Brickell et al. (2008), we generated two types of random weighted complete graphs. For type one graphs, for each edge $e$ we set $w(e) = 1$ with probability 0.8 and and $w(e) = 0$ with probability 0.2. For type two graphs, we let $w(e) \sim \mathcal{N}(0, 1)$. For both types of graphs, we ran both algorithms until the distance between the current iterate and its optimal decrease only solution (see Brickell et al. (2008)) was smaller than one. The computations were done on a machine with four physical cores with 13 GB of memory per core.

We can see from Figure 2 that as $n$ grows, our algorithm outperforms the algorithm from Brickell et al. (2008). We also see that our algorithm has less variability in its running time. Additionally, since we forget constraints, we are interested in the number of active constraints for these problems. From our experiments, we see that for type one graphs, our algorithm consistently returns $n^2/2$ constraints, and for type two graphs, consistently returns $n^2$ constraints.

In general, Brickell et al. (2008)'s algorithm works for $G = K_n$ only. For real-world data, we may not have full information about the interactions between data points; such relations may be

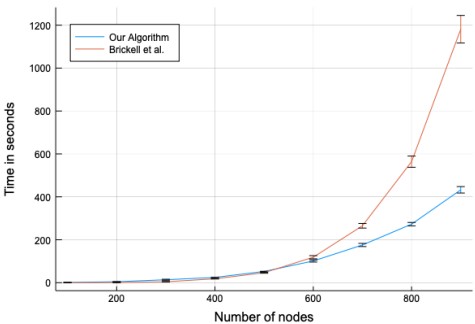 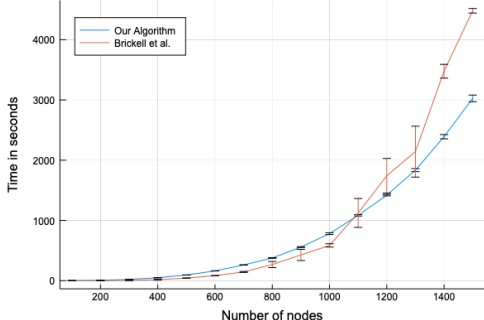

Figure 2: The red line is the mean running time over 5 samples for the algorithm from Brickell et al. (2008). The blue line is the running mean time for our algorithm. On the left we have the running times for type one graphs and on the right we have the running times for type two graphs.

sparse. Therefore, we do not restrict ourselves to finding the closest point in $\text{MET}_n$ when seeking a metric. Instead, we want to find the closest point in $\text{MET}(G)$ where $G$ is the graph representing the interactions for which we have information. As seen in Section 4.1.2 we can solve the problem when $G$ is sparse.

### 4.3 Metric learning

For the final problem, we solve a variant of the metric learning problem, ITML. To do a head to head comparison against Davis et al. (2007), we implemented their algorithm in Julia. As the algorithm from Davis et al. (2007) doesn't solve the complete linear program, only an approximation, we compare our algorithms on the quality of the solution.

For each data set, we uniformly at random choose $80\%$ of the data points to be the training set and the remaining to be the test set. We then let the similar pairs $S$ be pairs that had the same label and the dissimilar pairs $D$ be all of the other pairs. For the algorithm from Davis et al. (2007), as suggested in the paper, we randomly sampled $20c^2$ constraints, where $c$ is the number of different classes and ran the algorithm so that it performed about $10^6$ projections.

Because the algorithm from Davis et al. (2007) is also based on Bregman projections, to do a fair head to head comparison, we limit our selves to the same number of projections and, on each iteration, randomly sampled $2 \times 10^5$ constraints, $10^5$ from $S$ and $10^5$ from $D$. We then ran the algorithm for 10 iterations. Note when sampling constraints we allowed repetition. That is, if a constraint was sampled $k$ times during an iteration, we projected onto it $k$ times.

Table 3: Table comparing the testing accuracy of Project and Forget and ITML. Here the accuracy for ITML is the average over 100 trials.

| Algorithm | Banana | Ionsphere | Coil2000 | Letter | Penbased | Spambase | Texture |
|---|---|---|---|---|---|---|---|
| Ourt | **0.88679** | **0.90000** | **0.93842** | 0.90250 | 0.97408 | **0.93587** | **0.99909** |
| ITML | **0.88816** | 0.86985 | **0.93842** | **0.92852** | **0.99039** | 0.90713 | 0.98390 |

The accuracy reported in Table 3 is based on one run of our algorithm since we have a theoretical guarantee that we converge to the optimal solution. However, since there is a great amount of variability in the constraints selected for the algorithm from Davis et al. (2007), we ran their algorithm 100 times and report the mean accuracy. As it can be seen from the table, in half cases our algorithm has better test accuracy as we solve the complete linear program.

These results are interesting as both methods are guaranteed theoretically to converge to their optimal solution; the difference in accuracy is due to the difference in the set of constraints being considered. We conclude that in the cases ITML does better on average, it indicates that subsampling the set of constraints, rather than using all of constraints, leads to lower generalization error. To verify that this difference in performance highlighted a difference in problems and not in convergence of the

different algorithms, we ran PROJECT AND FORGET for $10^5$ iterations, where we sample $2 \times 10^3$ constraints per iteration and did not see any significant improvement in the accuracy.

## 5 FUTURE WORK

In conclusion we see that our algorithm PROJECT AND FORGET can be used to solve not only large scale versions of three metric constrained problems but also opens many avenues for future work.

The first direction is motivated by ITML. As we see in Table 3, sometimes it is better to select only a small subset of the constraints rather than computing with all of them. It is an interesting question whether we can determine what inherent properties of the data set result in this phenomenon. If we can answer this, we can reduce the number of constraints by several orders of magnitude.

Another avenue of work that this algorithm opens is in learning better, more general metrics. Many existing metric learning algorithms learn a linear transform of Euclidean space and then use this transformed space to obtain a more relevant metric. Using our technique, however, we can directly learn a Euclidean metric that optimizes the objective function which we could then embed into Euclidean space using multidimensional scaling or some other embedding technique. We are also no longer restricted to Euclidean metrics, but can learn hyperbolic and tree metrics as well, getting better metric representations of our data. With these techniques, we would gain a better understanding the space a data set lies in and a better, usable metric representation.

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

# 6 CONVEX PROGRAMMING

As we mentioned in main body of the paper, there is a much more general formulation of the metric constrained optimization problems and a correspondingly general variant of the algorithm. We present that setting below and detail the convergence results in the following section.

Given a strictly convex function $f$, and a finite family of half-space $\mathcal{H} = \{H_i\}$ such that $H_i = \{x : \langle a_i, x \rangle \leq b_i\}$, we want to find the unique point $x^* \in \bigcap_i H_i =: C$ that minimizes $f$. That is, if we set $A$ to be the matrix whose rows are $a_i$ and $b$ be the vector whose coordinates are $b_i$ we seek

$$\begin{aligned} \text{minimize} \quad & f(x) \\ \text{subject to} \quad & Ax \leq b \end{aligned} \tag{6.1}$$

We refer to each $H_i$ as a *constraint set* and $C$ as the *feasible region*. We shall assume that $C$ is not empty, i.e., there is at least one feasible point. In general, we can represent any half-space by a linear inequality.

In general, since $A$ is an extremely large matrix, computing $A$ or writing $A$ down is not computationally feasible for many large instances of the problem. Therefore, we access the constraint sets only through an oracle that has one of the two following separation properties.

**Property 1.** *$\mathcal{Q}$ is a deterministic separation oracle for a family of half-spaces $\mathcal{H}$, if there exists a positive, non-decreasing, continuous function $\phi$, with $\phi(0) = 0$, such that on input $x \in \mathbb{R}^d$, $\mathcal{Q}$ either certifies $x \in C$ or returns a list $\mathcal{L} \subset \mathcal{H}$ such that*

$$\max_{\tilde{C} \in \mathcal{L}} \text{dist}(x, \tilde{C}) \geq \phi(\text{dist}(x, C)).$$

**Property 2.** *$\mathcal{Q}$ is a random separation oracle for a family of half-spaces $\mathcal{H}$, if there exists some distribution $\mathcal{D}$ and a lower bound $\tau > 0$, such that on input $x \in \mathbb{R}^d$, $\mathcal{Q}$ returns a list $\mathcal{L} \subset \mathcal{H}$ such that*

$$\forall \tilde{H} \in \mathcal{H}, \; Pr_{\mathcal{D}}[\tilde{H} \in \mathcal{L}] \geq \tau.$$

For the random oracle, we do not need it to decide whether $x \in C$, so this oralce can be used to solve optimization problems over polytopes for which deciding membership is NP-hard.

In addition our method works for a rich class of functions known as Bregman functions.

**Definition 3.** *A function $f : \Lambda \to \mathbb{R}$ is called a Bregman function if there exists a non-empty convex set $S$ such that $\overline{S} \subset \Lambda$ and the following hold:*

- *(i) $f(x)$ is continuous, strictly convex on $\overline{S}$, and has continuous partial derivatives in $S$.*
- *(ii) For every $\alpha \in \mathbb{R}$, the partial level sets $L_1^f(y, \alpha) := \{x \in \overline{S} : D_f(x, y) \leq \alpha\}$ and $L_2^f(x, \alpha) := \{y \in S : D_f(x, y) \leq \alpha\}$ are bounded for all $x \in \overline{S}, y \in S$.*
- *(iii) If $y_n \in S$ and $\lim_{n \to \infty} y_n = y^*$, then $\lim_{n \to \infty} D_f(y^*, y_n) = 0$.*
- *(iv) If $y_n \in S$, $x_n \in \overline{S}$, $\lim_{n \to \infty} D_f(x_n, y_n) = 0$, $y_n \to y^*$, and $x_n$ is bounded, then $x_n \to y^*$.*

*We denote the family of Bregman functions by $\mathcal{B}(S)$. We refer to $S$ as the zone of the function and we take the closure of the $S$ to be the domain of $f$.*

**Definition 4.** *We say that a hyperplane $H_i$ is* strongly zone consistent *with a respect to a Bregman function $f$ and its zone $S$, if for all $y \in S$ and for all hyperplanes $H$, parallel to $H_i$ that lie in between $y$ and $H_i$, the Bregman projection of $y$ onto $H$ lies in $S$ instead of in $\overline{S}$.*

Next, we briefly discuss under what assumptions can we solve solve our optimization problem and what these assumptions mean.

**Assumption 1.** *$f(x)$ is a Bregman function.*

This class of function includes many natural objective functions, including $f(x) = -\sum_{i=1}^n x_i \log(x_i)$ with zone $S = \mathbb{R}_+^n$ (here $f$ is defined on the boundary of $S$ by taking the limit) and $f(x) = \frac{1}{p}\|x\|_p^p$ for $p \in (1, \infty)$. The $\ell_p$ norms for $p = 1, \infty$ are not Bregman functions but can be made Bregman functions by adding a quadratic term. That is, $f(x) = c^T x$ is a not Bregman function, but $c^T x + x^T Q x$ for any positive definite $Q$ is a Bregman function.

Additionally, strongly convex functions and Legendre functions are related to Bregman functions, but neither class implies the other. See Bauschke & Lewis (2000) for an example and a more in-depth discussion.

**Assumption 2.** *All hyperplanes in $\mathcal{H}$ are strongly zone consistent with respect to $f(x)$.*

This assumption is used to guarantee that when we do a projection the point we project onto lies within our domain. This is not too restrictive. For example, all hyperplanes are strongly zone consistent with respect to the objective functions $f(x) = 0.5\|x\|^2$ and $f(x) = -\sum_i x_i \log(x_i)$.

**Assumption 3.** $C$ *is non-empty.*

This is needed to make sure that the algorithm converges.

## 7 The general algorithms

To set the stage for subsequent discussions, we present the general structure of all of our algorithms. They are all iterative and, in general, are run until some convergence criterion has been met. The convergence criterion depends largely on the specific application for which the algorithm is tailored. For this reason, we postpone the discussion of the convergence criterion until the applications section.

Each iteration consists of three phases, as shown in algorithm 3. In the first phase, we query our oracle $\mathcal{Q}$ to obtain a list of constraints $L$. In the second phase, we merge $L^{(\nu)}$, the list of constraints we have been keeping tracking of up to the $\nu$th iteration, with $L$ and project onto each of the constraints in the list $L^{(\nu)} \cup L$ iteratively. Finally, in the third phase, we forget some constraints.

---

**Algorithm 3** General Algorithm.

1: **function** F($\mathcal{Q}, f$)
2:     $L^{(0)} = \emptyset$, $z^{(0)} = 0$. Initialize $x^{(0)}$ so that $\nabla f(x^{(0)}) = 0$.
3:     **while** Not Converged **do**
4:         $L = \mathcal{Q}(x^\nu)$
5:         $\tilde{L}^{(\nu+1)} = L^{(\nu)} \cup L$
6:         $x^{(\nu+1)} = \text{Project}(x^{(\nu)}, \tilde{L}^{(\nu+1)})$
7:         $L^{(\nu+1)} = \text{Forget}(\tilde{L}^{(\nu+1)})$
    **return** $x$

---

### 7.1 Project and Forget algorithms

The project and forget steps for algorithm are presented in Algorithm 4. Let us step through the code to understand the behavior of the algorithm. Let $H_i = \{x : \langle a_i, x \rangle \leq b_i$ be our constraint and $x$ our current iterate. The first thing we do is calculate $x^*$ and $\theta$. Here $x^*$ is the projection of $x$ onto the boundary of $H_i$ and $\theta$ is a "measure" of how far $x$ is from $x^*$. However, $\theta$ can be any real number. Let us discuss separately the cases $\theta$ positive and negative below.

It can be easily seen Censor & Zenios (1997) that $\theta$ is negative if and only if the constraint is violated. In this case, $c = \theta$. This is because (as we will see in proof) we will always maintain that $z_i \geq 0$. Then on line 5, we just compute the projection of $x$ onto $H_i$. Finally, since we corrected $x$ for this constraint, we add $|\theta|$ to $z_i$. Since each time we correct for $H_i$, we add to $z_i$, we see that $z_i$ stores how the total corrections made for $H_i$.

On the other hand, if $\theta$ is positive, this means that this constraint is satisfied. In this case, if we also have that $z_i$ is positive, i.e., we have corrected for $H_i$ before, then we have over compensated for this constraint. Hence we must do undo some of the corrections. Thus, if $c = z_i$ then we undo all of the corrections and $z_i$ is now 0. Otherwise if $c = \theta$ we only undo part of the correction.

The FORGET step is relatively easy, given a constraint $H_i$ we just need to check if $z_i = 0$. If $z_i = 0$, then this means we have not done any net corrections for this constraint. Thus, we can forget about it; i.e., delete it from $L^{(\nu)}$.

---

**Algorithm 4** Project and Forget algorithms.

---

1: **function** PROJECT($x, z, L$)
2:     **for** $H_i = \{y : \langle a_i, y \rangle = b_i\} \in L$ **do**
3:         Find $x^*, \theta$ by solving $\nabla f(x^*) - \nabla f(x) = \theta a_i$ and $x^* \in H_i$
4:         $c_i = \min(z_i, \theta)$
5:         $x \leftarrow$ such that $\nabla f(x^{n+1}) - \nabla f(x) = c_i a_i$
6:         $z_i \leftarrow z_i - c_i$
        **return** $x, z$
7: **function** FORGET($x, z, L$)
8:     **for** $H_i = \{x : \langle a_i, x \rangle = b_i\} \in L$ **do**
9:         **if** $z_i == 0$ **then** Forget $H_i$
        **return** $L$

---

In general, calculating the Bregman projection (line 3) cannot be done exactly. See Dhillon & Tropp (2007) for a general method to perform the calculation on line 3 and for an analytic formula for when $f$ is a quadratic function.

## 7.2 CONVERGENCE ANALYSIS

Now that we have specified our algorithm, we need to establish a few crucial theoretical properties. The first is that under assumptions 1,2,3 our algorithm is guaranteed to converge to the global optimum solution. In fact, we also show that asymptotically, our error decreases at an exponential rate. These main theoretical results can be summarized by the following theorem.

**Theorem 3.** *If $f \in \mathcal{B}(S)$, $H_i$ are strongly zone consistent with respect to $f$, and $\exists x^0 \in S$ such that $\nabla f(x^0) = 0$, then*

1. *If the oracle $\mathcal{Q}$ satisfies property 1 (property 2), then any sequence $x^n$ produced by the above algorithm converges (with probability 1) to the optimal solution of problem 6.1.*

2. *If $x^*$ is the optimal solution, $f$ is twice differentiable at $x^*$, and the Hessian $H := Hf(x^*)$ is positive semidefinite, then there exists $\rho \in (0, 1)$ such that*

$$\lim_{\nu \to \infty} \frac{\|x^* - x^{\nu+1}\|_H}{\|x^* - x^\nu\|_H} \leq \rho \tag{7.1}$$

*where $\|y\|_H^2 = y^T H y$. In the case when we have an oracle that satisfies property 2, the limit in 7.1 holds with probability 1.*

*Proof.* The proof of this theorem has been moved to the supplementary material section. $\square$

The proof of Theorem 3 also establishes another important theoretical property.

**Proposition 3.** *If $a_i$ is an inactive constraint, then (with probability 1) $z_i^\nu = 0$ for the tail of the sequence.*

**Corollary 2.** *Under the assumptions for part (2) of Theorem 3, we have that the sequence $z^n \to z^*$ also converges.*

These properties are important as they permit the following interpretation of our algorithm. The algorithm spends the initial few iterations identifying the active constraints from amongst a large number of constraints. The algorithm then spends the remainder of the iterations finding the optimal solution with respect to these constraints. This ability to find the set of active constraints is one of the main advantages of our algorithm.

## 7.3 THE METRIC CONSTRAINED VERSIONS

First we can easily see that function METRIC VIOLATION in Algorithm 1 is an oracle that satisfies Property 1

**Proposition 4.** *Function* METRIC VIOLATION *in Algorithm 1 is an* $\Theta(n^2 \log(n) + n|E|)$ *oracle that has Property 1*

*Proof.* The first step in METRIC VIOLATION is to calculate the shortest distance between all pairs of nodes. This can be done using Dijkstra's algorithm in $\Theta(n^2 \log(n) + n|E|)$ time. Then if the shortest path between any adjacent pair of vertices is not the edge connecting them, then the algorithm has found a violated cycle inequality. Note that if no such path exists, then all cycle inequalities have been satisfied. Hence our point is within the metric polytope. Thus, we have an oracle that separates the polytope. However, we want an oracle that satisfies property 1.

Given a point $x$ and a hyperplane $H_{C,e}$, defined by some cycle $C$ and an edge $e$, the deficit of this constraint is given by the following formula.

$$d(C, e) = x(e) - \sum_{\tilde{e} \in C, \tilde{e} \neq e} x(\tilde{e})$$

If this quantity is positive, then $x$ violates this constraint. In this case, the distance from $x$ to this constraint is $\frac{d(C,e)}{|C|}$. While the above oracle does not find the cycle $C$ and edge $e$ that maximize this distance, it does find the cycle $C$ and edge $e$ that maximize $d(C, e)$. Since $|C| \in [1, n]$, if we let $\phi(x) = x/n$, we see that oracle satisfies property 1. $\qquad\square$

Similarly we can see that uniformly randomly sampling constraints is an oracle that satisfies Property 2. Thus we can see that both algorithms presented in the main text are special cases of the above algorithm.

## 7.4  TRULY STOCHASTIC VARIANT

Now Algorithm 4 can be used with an oracle that satisfies property 2. However, this is not a completely stochastic algorithm. We still have to keep track of the constraints that we have seen and carefully pick which constraints to forget. Nevertheless, we can modify our forget step to forget all constraints and obtain a truly stochastic version of the algorithm. In this version, at each iteration, we choose a random set of constraint and project onto these constraints only, independently of what constraints were used in previous iterations. We cannot, however, forget the values of the dual variables. In this case, we have the following convergence result.

**Theorem 4.** *If* $f \in \mathcal{B}(S)$, $H_i$ *are strongly zone consistent with respect to* $f$, *and* $\exists x^0 \in S$ *such that* $\nabla f(x^0) = 0$, *then with probability* $1$ *any sequence* $x^n$ *produced by the above truly stochastic algorithm converges to the optimal solution of problem 6.1. Furthermore, if* $x^*$ *is the optimal solution, $f$ is twice differentiable at* $x^*$, *and the Hessian* $H := Hf(x^*)$ *is positive semidefinite, then there exists* $\rho \in (0, 1)$ *such that with probability* $1$,

$$\liminf_{\nu \to \infty} \frac{\|x^* - x^{\nu+1}\|_H}{\|x^* - x^\nu\|_H} \leq \rho.$$

This version of our algorithm is very similar to the algorithms presented Nedić (2011); Wang et al. (2015). The major difference being that we do not need to a gradient descent step. Instead we maintain the KKT conditions, by keeping track of the dual variables and doing dual corrections.

However, in practice using 4 with the random oracle tends to produce better results. This is because by not forgetting the active constraints that we have seen, instead of hoping that we sample them, we speed up convergence significantly.

## 8  CONVERGENCE RESULTS

In this section, we present the proofs of Theorems 3 and 4. Because the proof of Theorem 3 is quite technical and involves two different types of separation oracles, we split it into several parts. In Subsections 8.1 and 8.2, we prove the first part of Theorem 3 for separation oracles with property 1 and 2, respectively. In Subsection 8.3, we prove the second part of Theorem 3 (also subdividing this proof into several cases). Finally, in Subsection 8.4 we prove Theorem 4, noting only the changes necessary from the proof of Theorem 3.

## 8.1 Proof of part 1 of Theorem 3 for oracles that satisfy property 1

We remind the reader of the notation established in Section 2. The vector of variables over which we optimize is $x$, $f$ is the objective function, $H_i = \{y : \langle y, a_i \rangle = b_i\}$ are the hyperplanes that lie on the boundaries of the half-space constraints, $L$ is the Lagrangian, $z$ is the dual variable, $A$ is the matrix with rows given by $a_i$, and $b$ is the vector with rows $b_i$.

Next, we clarify the indexing of the variables. Algorithm 3 has three steps per iteration and during the PROJECT step there are multiple projections. When we want to refer to a variable after the $\nu$th iteration, it will have a superscript with a $\nu$. When we refer to a variable after the $n(i, k)$th projection, we use the superscript $n(i, k)$. Finally, before the $n$th projection, $i(n)$ will represent the index of the hyperplane onto which we project.

Finally, let $R$ be the maximum number of constraints that our oracle $\mathcal{Q}$ returns. This is clearly upper bounded by the total number of constraints, which we have assumed is finite. We are now ready to prove the first part of Theorem 3.

**Theorem 3.** *(Part 1) If $f \in \mathcal{B}(S)$, $H_i$ are strongly zone consistent with respect to $f$, $\exists x^0 \in S$, such that $\nabla f(x^0) = 0$, and the oracle $\mathcal{Q}$ satisfies property 1, then any sequence $x^n$ produced by Algorithm 3 converges to optimal solution of problem 6.1.*

*Proof.* The proof of this theorem is an adaptation of the proof of convergence for the traditional Bregman method that is presented in Censor & Zenios (1997) whose proof entails the following four steps. The main difference between Censor and Zenios' proof and ours is that of the last two steps. We present the entire proof, however, for completeness. To that end, we show that

**Step 1.** the KKT condition, $\nabla f(x) = \nabla f(x^0) - A^T z$, is always maintained,

**Step 2.** the sequence $x^n$ is bounded and it has at least one accumulation point,

**Step 3.** any accumulation point of $x^n$ is feasible (i.e., is in $C$), and

**Step 4.** any accumulation point is the optimal solution.

**Step 1.** The KKT condition, $\nabla f(x) = \nabla f(x^0) - A^T z$, is always maintained.

We show by induction that for all $n$, $\nabla f(x^n) = -A^T z^n$. In the base case, $z = 0$, thus, $\nabla f(x^0) = 0 = -A^T z^0$. Assume the result holds for iteration $n$, then

$$\nabla f(x^{n+1}) = \nabla f(x^n) + c^n a_{i(n)} = -A^T z^n + A^T c^n e_{i(n)} = -A^T (z^n - c^n e_{i(n)}) = -A^T z^{n+1}$$

We know that $c^n \leq z^n_{i(n)}$; therefore, we maintain $z^{n+1} \geq 0$ as well.

**Step 2.** The sequence $x^n$ is bounded and has an accumulation point.

To show that $x^n$ is a bounded, we first show that $\left(L(x^n, z^n)\right)_n$ is a monotonically increasing sequence bounded from above. This observation results from the following string of equalities:

$$
\begin{aligned}
L(x^{n+1}, z^{n+1}) - L(x^n, z^n) &= f(x^{n+1}) - f(x^n) + \langle z^{n+1}, Ax^{n+1} - b \rangle - \langle z^n, Ax^n - b \rangle \\
&= f(x^{n+1}) - f(x^n) + \langle A^T z^{n+1}, x^{n+1} \rangle - \langle A^T z^n, x^n \rangle - \langle z^{n+1} - z^n, b \rangle \\
&= f(x^{n+1}) - f(x^n) - \langle \nabla f(x^{n+1}), x^{n+1} \rangle + \langle \nabla f(x^n), x^n \rangle + \langle c^n e_{i(n)}, b \rangle \\
&= f(x^{n+1}) - f(x^n) - \langle \nabla f(x^n) + c^n a_{i(n)}, x^{n+1} \rangle + \langle \nabla f(x^n), x^n \rangle + c^n b_{i(n)} \\
&= f(x^{n+1}) - f(x^n) - \langle \nabla f(x^n), x^{n+1} - x^n \rangle - \langle c^n a_{i(n)}, x^{n+1} \rangle + c^n b_{i(n)} \\
&= \underbrace{D_f(x^{n+1}, x^n)}_{(1)} + \underbrace{c^n (b_{i(n)} - \langle a_{i(n)}, x^{n+1} \rangle)}_{(2)}
\end{aligned}
$$

Next, we show that both terms (1) and (2) are non-negative. We know that $D_f$ is always non-negative so we only need to consider term (2). There are two cases: (i) if $c^n = \theta^n$, then $x^{n+1} \in H_{i(n)}$ and

$b_{i(n)} - \langle a_{i(n)}, x^{n+1} \rangle = 0$. On the other hand, (ii) if $c^n = z^n_{i(n)}$, then $b_{i(n)} - \langle a_{i(n)}, x^{n+1} \rangle \geq 0$ and $c^n \geq 0$. We can conclude that the difference between successive terms of $L(x^n, z^n)$ is always non-negative and, hence, it is an increasing sequence.

To bound the sequence, let $y$ be a feasible point (i.e., $Ay \leq b$). (Note that this is the only place we use the assumption that the feasible set is not empty.) Then

$$
\begin{aligned}
D_f(y, x^n) &= f(y) - f(x^n) - \langle \nabla f(x^n), y - x^n \rangle \\
&= f(y) - f(x^n) + \langle z^n, Ay - Ax^n \rangle \\
&\leq f(y) - f(x^n) + \langle z^n, b - Ax^n \rangle.
\end{aligned}
$$

Rearranging terms in the inequality, we obtain a bound on the sequence $L(x^n z^n)$ from above:

$$
L(x^n, z^n) = f(x^n) + \langle z^n, Ax^n - b \rangle \leq f(y) - D_f(y, x^n) \leq f(y).
$$

Since the sequence $(L(x^n, z^n))_{n \in \mathbb{N}}$ is increasing and bounded, it is a convergent sequence and the difference between successive terms of the sequence goes to 0. Therefore,

$$
\lim_{n \to \infty} D_f(x^{n+1}, x^n) = 0.
$$

From the previous inequality we also have that

$$
D_f(y, x^n) \leq f(y) - L(x^n, z^n) \leq f(y) - L(x^0, z^0) =: \alpha.
$$

Using part (ii) of the definition of a Bregman function, we see that $L^f_2(y, \alpha)$ is bounded and since $(x^n)_{n \in \mathbb{N}} \in L^f_2(y, \alpha)$, $x^n$ is a bounded sequence with an accumulation point.

**Step 3.** Any accumulation point $x^*$ of $x^n$ is feasible (i.e., is in $C$).

This is the only step in which we use the fact that our oracle satisfies property 1. Let $x^*$ be some accumulation point for $x^n$ and assume for the sake of contradiction that $Ax^* \not\leq b$. Let $\tilde{A}, \tilde{b}$ be the maximal set of constraints that $x^*$ does satisfy; i.e.,

$$
\tilde{A} x^* \leq \tilde{b}.
$$

Let $(x^{n_k})$ be a subsequence such that $x^{n_k} \to x^*$ and $H$ be a constraint that $x^*$ violates. Define $\epsilon$ as

$$
\epsilon := \phi(d(x^*, H)) > 0. \tag{8.1}
$$

Because $x^n$ is bounded, $x^{n_k}$ is a convergent subsequence $x^{n_k} \to x^*$, and $D_f(x^{n+1}, x^n) \to 0$, by equation 6.48 from Censor & Zenios (1997) we see that for any $t$,

$$
x^{n_k + t} \to x^*.
$$

In particular, the proposition holds for all $t \leq 2|\tilde{A}| + 2 =: T$.

Let us consider an augmented subsequence $x^{n_k}, x^{n_k+1}, \ldots, x^{n_k+T}$, i.e., add in extra terms. Note that if $n_{k+1} - n_k \to \infty$, then this augmented sequence is not the entire sequence. We want to show that infinitely many of the terms in our augmented sequence satisfy a constraint *not* in $\tilde{A}$. Should this hold, then because we have only finitely many constraints, there exists at least one single constraint $\tilde{a}$ that is *not* in $\tilde{A}$, such that infinitely many terms of the augmented sequence all satisfy the single constraint $\tilde{a}$. Finally, because our augmented sequence converges to $x^*$ and we are only looking at closed constraints, we must have that $x^*$ also satisfies the constraint $\tilde{a}$. Thus, we would arrive at a contradiction of the maximality of $\tilde{A}$ and $x^*$ would have to be in the feasible region.

To see that infinitely many of the terms in our augmented sequence satisfy a constraint *not* in $\tilde{A}$, let $\nu_k$ be the iteration in which the $n_k$th projection takes place. Note that we can assume without loss of generality that in any iteration, we project onto any constraint at most once. If this were not the case and we projected onto constraints more often, we would simply change the value of $T$ to reflect this larger number of projections. Therefore, we have two possibilities for which iteration the $n_k + |\tilde{A}| + 1$st projection takes place and we consider each case below.

**Case 1:** The $n_k + |\tilde{A}| + 1$st projection, infinitely often, takes places in $\nu_k$th iteration. Since we project onto each constraint at most once, one of the projections between the $n_k$ and $n_k + |\tilde{A}| + 1$st

projection must be onto a hyperplane defined by a constraint *not* represented in $\tilde{A}$ and amongst the terms $x^{n_k}, x^{n_k+1}, \ldots, x^{n_k+|\tilde{A}|+1}$, we must have a term that satisfies a constraint *not* in $\tilde{A}$ infinitely often.

**Case 2:** The $n_k + |\tilde{A}| + 1$st projection, infinitely often, takes place in $\nu_k + 1$st iteration or later.

If this projection happens in $\nu_k + 1$st iteration, consider the iteration in which we do the $n_k + T$th projection. If this projection also takes place in the $\nu_k + 1$st iteration, then we have done at least $|\tilde{A}| + 1$ projections in the $\nu_k + 1$st iteration. Hence, amongst $x^{n_k+|\tilde{A}|+1}, \ldots, x^{n_k+T}$, we must have a term that satisfies a constraint *not* in $\tilde{A}$.

If the $n_k + |\tilde{A}| + 1$st or the $n_k + T$th projection happens in the $\nu_k + 2$nd iteration or later, then between the $n_k$th and the $n_k + T$th projection, we must have projected onto all constraints returned by oracle in the $\nu_k + 1$st iteration. Therefore, we must have projected onto some hyperplane defined by $\hat{a}$ (for some constraint $\hat{C}$) such that

$$\hat{d}^{n_k} := d(x^{\nu_k+1}, \hat{C}) \geq \phi(d(x^{\nu_k+1}, C)).$$

Then there exists a sufficiently small $\delta > 0$, depending on $\tilde{A}, \tilde{b}, x^*$, such that if $\|y - x^*\| \leq \delta$, then

$$\tilde{A}y \leq \tilde{b} + \frac{\epsilon}{2}\mathbb{1},$$

where $\mathbb{1}$ is vector of all ones.

Since our augmented sequence converges to $x^*$, we know that there exists a $K$, such that for all $k \geq K$ and $t \leq T$, $\|x^{n_k+t} - x^*\| < \delta$. That is, for all $k \geq K$ and $t \leq T$,

$$\tilde{A}x^{n_k+t} \leq \tilde{b} + \frac{\epsilon}{2}\mathbb{1}. \tag{8.2}$$

Note $x^{\nu_k+1}$ is within our augmented sequence so if $\hat{a}$ is infinitely often in $\tilde{A}$, by equation 8.2, we have that infinitely often

$$\frac{\epsilon}{2} \geq \hat{d}^{n_k}.$$

Finally, because the augmented sequence converges to $x^*$,

$$\epsilon = \phi(d(x^*, H)) \leq \phi(d(x^*, C)) = \lim_{k\to\infty} \phi(d(x^{\nu_k+1}, C)) \leq \lim_{k\to\infty} \hat{d}^{n_k} \leq \frac{\epsilon}{2}.$$

The first inequality follows from the fact that $\phi$ is non-decreasing. Therefore, $\hat{a}$ is *not* in $\tilde{A}$ infinitely often and amongst $x^{n_k}, x^{n_k+1}, \ldots, x^{n_k+T}$, we must have a term that satisfies a constraint *not* in $\tilde{A}$ infinitely often. Thus, there is a constraint $\tilde{a}$ not in $\tilde{A}$ that is satisfied by infinitely terms of our augmented sequence and we have a contradiction.

**Step 4.** Optimality of accumulation point.

Because we have established the feasibility of all accumulation points, we show next that any accumulation point $x^{n_k} \to x^*$ is optimal.

First, we show that there exists an $N$, such that for any $k \geq N$, and for any $a_i$ such that

$$\langle a_i, x^* \rangle < b_i,$$

we have $z_i^{n_k} = 0$. To do so, we assume for the sake of contradiction that for some $a_i$, our sequence $z_i^{n_k}$ is infinitely often not 0. The algorithm then projects onto this constraint infinitely often. Therefore, the point $x^{n_k}$ lies on the hyperplane defined by $a_i, b_i$ infinitely often. Thus, the limit point $x^*$ must lie on this hyperplane as well and we have a contradiction.

Now we know that for any constraint $a_i$, we either have that $\langle a_i, x^* \rangle = b_i$ or we have that $z_i^{n_k} = 0$ for the tail of the sequence. Thus, for sufficiently large $k$,

$$\langle z^{n_k}, Ax^{n_k} - b \rangle = \langle A^T z^{n_k}, x^{n_k} - x^* \rangle = \langle -\nabla f(x^{n_k}), x^{n_k} - x^* \rangle = D_f(x^*, x^{n_k}) - f(x^*) + f(x^{n_k}).$$

Next, by part (iii) of the definition of a Bregman function,

$$\lim_{k\to\infty} D_f(x^*, x^{n_k}) = 0.$$

Finally,

$$\lim_{k \to \infty} L(x^k, z^k) = \lim_{k \to \infty} f(x^{n_k}) + \langle z^{n_k}, Ax^{n_k} - b \rangle = f(x^*).$$

We also know that $L(x^k, z^k) \leq f(y)$ for any feasible $y$. Thus, $f(x^*) \leq f(y)$. Hence $x^*$ is an optimal solution. Now since $f$ is strictly convex, this optimal point is unique. Therefore, we have that $(x^n)_{n \in \mathbb{N}}$ has only one accumulation point and $x^n \to x^*$. $\qquad\square$

An important fact consequence of this proof is the following proposition:

**Proposition 3.** *If $a_i$ is an inactive constraint, then there exists a $N$, such that for all $n \geq N$, we have that $z_i^n = 0$. That is, after some finite time, we never project onto inactive constraints ever again.*

## 8.2 PROOF OF PART 1 OF THEOREM 3 FOR ORACLES THAT SATISFY PROPERTY 2

In this subsection, we prove part 1 of Theorem 3 for oracles that satisfy property 2. We make note of the key ideas in this proof as they are useful in the proof of the truly stochastic variant. To be precise, we prove:

**Theorem 1.** *(Part 1) If $f \in \mathcal{B}(S)$, $H_i$ are strongly zone consistent with respect to $f$, $\exists x^0 \in S$, such that $\nabla f(x^0) = 0$, and the oracle $\mathcal{Q}$ satisfies property 2, then with probability 1, any sequence $x^n$ produced by Algorithm 3 converges to optimal solution of problem 6.1.*

*Proof.* Assume that we have an oracle that satisfies property 2. A careful reading of the previous proof shows that if we switch out an oracle with property 1 for an oracle with property 2, then we only need to adjust step 3 of our proof. The crucial part of that step was showing that for our augmented sequence, we had infinitely many terms that satisfied a constraint *not* in $\tilde{A}$. We make the following adjustments to our analysis.

Let $\nu_k$ be the iteration in which the $n_k$th projection takes place. In the previous proof, we used the property of the oracle only when the $n_k + T$th projection took place in the $\nu_k + 2$nd iteration or a later iteration. In this case, the augmented sequence encompasses all of the $\nu_k + 1$st iteration infinitely often.

Let us choose a constraint $\hat{a}$ that is not satisfied by $x^*$. Because the oracle satisfies property 2, for each iteration $\nu_k + 1$, our oracle returns $\hat{a}$ with probability at least $\tau > 0$. By the Borel Cantelli Lemma, we know that during the selected iterations, the constraint $\hat{a}$ is, with probability one, returned infinitely often by our oracle. Thus, our augmented sequence satisfies this constraint with probability 1 and $x^*$ lies in the feasible region with probability 1. $\qquad\square$

A direct consequence of this proof is the proof of the probabilistic version of Proposition 3.

**Proposition 3.** *With probability 1, we project onto inactive constraints a finite number of times.*

## 8.3 PROOF OF PART 2 OF THEOREM 3

The discussions in Iusem & De Pierro (1990); Iusem (1991) almost directly apply to that for our algorithm. For completeness, we present it along with the necessary modifications. As with the traditional Bregman algorithm, we first present the case when $f(x)$ is quadratic. That is,

$$f(x) = r + s^T \cdot x + \frac{1}{2} x^T H x$$

where $H$ is a positive definite matrix. In this case, it is easy to see that

$$D_f(x, y) = \|x - y\|_H^2 := (x - y)^T H (x - y).$$

### 8.3.1 PROOF OF PART 2 OF THEOREM 3—QUADRATIC CASE

In this section, we will prove the following variation of Theorem 3.

**Theorem 3.** *If $f$ is a strictly convex quadratic function, $H_i$ are strongly zone consistent with respect to $f$, $x^0 = H^{-1}s \in S$, and the oracle $Q$ satisfies either property 1 or 2, then there exists $\rho \in (0,1)$ such that*

$$\lim_{\nu \to \infty} \frac{\|x^* - x^{\nu+1}\|_H}{\|x^* - x^\nu\|_H} \leq \rho \tag{8.3}$$

*where $\|y\|_H^2 = y^T H y$. In the case when we have an oracle that satisfies property 2, the limit in 8.3 holds with probability 1.*

We establish some notation ahead of our lemmas. Let $I$ be the set of all active constraints. That is, if $x^*$ is the optimal solution then

$$I = \{i : \langle a_i, x^* \rangle = b_i\}.$$

Let $S$ be the set of all $x$ that satisfy these constraints (namely $S = \{x : \forall i \in I, \langle a_i, x \rangle = b_i\}$). Let $H_x$ be the hyperplane, such that $H_x$ represents the constraint in $I$ that is furthest from $x$. Define

$$\mu = \inf_{x \notin S} \frac{d(x, H_x)}{d(x, S)}.$$

By Iusem & De Pierro (1990), we know that $\mu > 0$. Let $U$ be the set of all optimal dual variables $z$; i.e., $U = \{z : \nabla f(x^*) = -A^T z\}$ and let $I_\nu = \{i : z_i^{\nu+1} \neq 0\}$.

Next, we present a few preliminary lemmas. These lemmas exist in some form or another in Iusem (1991); Iusem & De Pierro (1990) and we present them suitably modified for our purpose. These lemmas require the following set of assumptions about an iteration $\nu$:

1. $\forall i \notin I, z_i^\nu = 0$;
2. for all $i \notin I$, we do not project onto this constraint in the $\nu$th iteration; and,
3. there exists $z \in U$, such that for all $i \notin I_\nu, z_i = 0$.

**Lemma 1.** *Let $x^*$ be the optimal solution for an instance of problem 6.1. For any sequence $x^n \to x^*$ such that $x^n, z^n$ maintain the KKT conditions, there exists an $M$, such that for all $\nu \geq M$, there exists a $z \in U$, such that for all $i \notin I_\nu$, we have that $z_i = 0$.*

*Proof.* Let $V_\nu = \{z : \forall i \notin I_\nu, z_i = 0\}$. Then assume, for the sake of contradiction, that the result is false. That is, there is a sequence $\nu_k$ such that $V_{\nu_k} \cap U = \emptyset$. Then since there finitely many different $I_\nu$ (hence finitely many $V_\nu$), we have that one of these must occur infinitely often. Thus, by taking an appropriate subsequence, we assume, without loss of generality, that $I_{\nu_k}$ are all equal. Let $V = V_{\nu_k}$ and obtain $V \cap U = \emptyset$.

Since $V$ is a closed subspace, $U$ is a closed set, and $V \cap U = \emptyset$, we must have that $d(V, U) > 0$. But $z^{\nu_k+1} \in V$ and so

$$d(z^{\nu_k+C}, U) \geq d(V, U) > 0.$$

Since $x^\nu \to x^*$ and we maintain the KKT conditions, we have that for any $z \in U$,

$$A^T z^\nu = -\nabla f(x^\nu) \to -\nabla f(x^*) = A^T z.$$

Thus $d(z^\nu, U) \to 0$ which is a contradiction. $\square$

**Lemma 2.** *For any sequence $x^n \to x^*$, if for a given $\nu$, we have that the sequence satisfies assumptions (1) and (2), then*

$$\|x^{\nu+1} - x^*\|_Q^2 \leq \|x^\nu - x^*\|_Q^2 - \sum_{n=k}^K \|x^{n+1} - x^n\|_Q^2$$

*where $k$ and $K$ are the indices of the first and last projection that take place in the $\nu$th iteration.*

*Proof.* This Lemma is simply a statement about Bregman projections and so its proof requires no modification. $\square$

Before we proceed, we introduce additional notation. Let $A_{I_\nu}, b_{i_\nu}$ be the submatrix of $A, b$ with rows from $I_\nu$ and

$$S_\nu = \{x : A_{I_{\nu_k}} x = b_{I_{\nu_k}}\}.$$

**Lemma 3.** *For any sequence such that $x^n \to x^*$, if for a given $\nu$, we have that it satisfies assumptions (1), (2), and (3), then we have that $\|x^{\nu+1} - x^*\|_Q = d(x^{\nu+1}, S_\nu)$.*

*Proof.* Consider the constrained problem

$$\min_{x \in S_\nu} \|x^{\nu+1} - x\|_Q^2 \tag{8.4}$$

Then sufficient conditions for a pair $(x, z_{I_\nu})$ to be optimal for this problem are

$$A_{I_\nu} x = b_{I_\nu} \text{ and } x = x^{\nu+1} - Q^{-1} A_{I_\nu}^T z_{I_\nu}$$

By Proposition 3, we see that since $x^*$ is solution to problem 6.1, we have that $A_{I_\nu} x^* = b_{I_\nu}$. Then by assumptions and the manner in which we do projections, we have that there exists $z \in U$, such that for all $i \notin I_\nu$, $z_i = 0$ and

$$x^* = x^{\nu+1} - Q^{-1} A^T (z^{\nu+1} - z)$$

Then since $z_i^{\nu+1} = 0$ for all $i \notin I_\nu$, we have that

$$x^* = x^{\nu+1} - Q^{-1} A_{I_\nu}^T (z_{I_\nu}^{\nu+1} - z_{I_\nu})$$

Thus, $x^*$ is the optimal solution to 8.4. $\square$

Next for $x \notin S_\nu$, let $H_x^\nu$ be the hyperplane of that is furthest from $x$ and define

$$\mu_\nu = \inf_{x \notin S_\nu} \frac{d(x, H_x^\nu)}{d(x, S_\nu)}$$

Now we are ready to prove the following theorem.

**Theorem 5.** *Let $x^*$ is the optimal solution to problem 6.1. Then given $\nu$ that satisfies assumptions (1), (2), and (3), we have that*

$$\|x^{\nu+1} - x^*\|_Q^2 \le \frac{L}{L + \mu^2} \|x^\nu - x^*\|_Q^2$$

*where $L$ is the number of projections that happened in $\nu$th iteration.*

*Proof.* By Lemma 3, for any such $\nu$ we have that $x^{\nu+1} \notin S_\nu$ (or we have converged already). Suppose constraint $j \in I_\nu$ defines the hyperplane $H_{x^{\nu+1}}^\nu$. Then by Lemma 3 and definitions of $\mu_\nu, \mu$ we have the following inequality.

$$\|x^{\nu+1} - x^*\| = d(x^{\nu+1}, S_\nu)$$
$$\le \frac{1}{\mu_\nu} d(x^{\nu+1}, H_{x^{\nu+1}}^\nu)$$
$$\le \frac{1}{\mu} d(x^{\nu+1}, H_{x^{\nu+1}}^\nu)$$

Now since $I_\nu = \{i : z_i^{\nu+1} \ne 0\}$, we know that during the $\nu$th iteration we must have projected onto $H_{x^{\nu+1}}^\nu$. Note that this is the only place in the proof where we need the fact that we remember old constraints. Let us say that this happens during the $r$th projection of the $\nu$th iteration.

Note by assumption, we satisfy the assumptions of Lemma 2. Let $y^r, y^{\nu+1}$ be the projections of $x^r, x^{\nu+1}$ onto $H_{x^{\nu+1}}^\nu$. Then we see that

$$d(x^{\nu+1}, H_{x^{\nu+1}}^{\nu})^2 = \|y^{\nu+1} - x^{\nu+1}\|_Q^2$$

$$\leq \|y^r - x^{\nu+1}\|_Q^2 \qquad\qquad [y^{\nu+1} \text{ by def is the closest point}]$$

$$\leq \left( \|y^r - x^{r+1}\|_Q + \sum_{i=r+1}^{L} \|x^i - x^{i+1}\|_Q \right)^2 \qquad [\text{Triangle inequality}]$$

$$= \left( \sum_{i=r+1}^{L} \|x^i - x^{i+1}\|_Q \right)^2$$

$$\leq \left( \sum_{i=0}^{L} \|x^i - x^{i+1}\|_Q \right)^2$$

$$\leq L \sum_{i=0}^{L} \|x^i - x^{i+1}\|_Q^2 \qquad\qquad [\text{ Cauchy Schwarz}]$$

$$\leq L \left( \|x^\nu - x^*\|_Q^2 - \|x^{\nu+1} - x^*\|_Q^2 \right) \qquad [\text{Lemma 2}]$$

Thus, we get that

$$\mu^2 \|x^{\nu+1} x^*\|^2 \leq d(x^{\nu+1}, H_{x^{\nu+1}}^{\nu})^2 \leq L \left( \|x^\nu - x^*\|^2 - \|x^{\nu+1} - x^*\|_Q^2 \right)$$

Rearranging, we get that

$$\|x^{\nu+1} - x^*\|_Q^2 \leq \frac{L}{L + \mu^2} \|x^\nu - x^*\|_Q^2.$$

$\square$

As a corollary to the above theorem, we have that algorithm 1 converges linearly.

**Theorem 3.** *If $f$ is a strictly convex quadratic function, $H_i$ are strongly zone consistent with respect to $f$, $x^0 = H^{-1}s \in S$, and the oracle $\mathcal{Q}$ satisfies either property 1 or 2, then there exists $\rho \in (0,1)$ such that*

$$\lim_{\nu \to \infty} \frac{\|x^* - x^{\nu+1}\|_H}{\|x^* - x^\nu\|_H} \leq \rho \qquad\qquad (8.3)$$

*where $\|y\|_H^2 = y^T H y$. In the case when we have an oracle that satisfies property 2, the limit in 8.3 holds with probability 1.*

*Proof.* Using Proposition 3, Lemma 1, and that we have finitely many constraints, we see that if $\nu$ is large enough, the assumptions for Theorem 5 are satisfied. Taking the limit gives us the needed result.

In the case when we have an oracle that satisfies property 2, consider the product space of all possible sequences of hyperplanes returned by our oracle. In this product space, we see that with probability 1, we generate a sequence of hyperplanes, such that algorithm 1 converges. For any such sequence of hyperplanes, we have that 8.3 holds. Thus, the limit in 8.3 holds with probability 1 for random separation oracles. $\square$

### 8.3.2 PROOF OF PART 2 OF THEOREM 3—GENERAL

The rate of convergence for the general Bregman method was established in Iusem (1991). To show this, let $\tilde{f}$ be the 2nd degree Taylor polynomial of $f$ centered at the optimal solution $x^*$.

$$\tilde{f}(x) = f(x^*) + \nabla f(x^*)^T \cdot x + \frac{1}{2} x^T \cdot \nabla^2 f(x^*) \cdot x$$

For notational convenience, let $H$ be the Hessian of $f$ at $x^*$. Then we can see that if replace $f$ with $\tilde{f}$ in 6.1 then the optimal solution does not change. Thus, if had access to $\tilde{f}$ and could use this function to do our projections, then from the quadratic case we have our result.

Thus, to get the general result, if $x^\nu$ is our standard iterate and $\tilde{x}^\nu$ is the iterate produced by using $\tilde{f}$ instead of $f$, then Iusem (1991) shows that $\|x^\nu - \tilde{x}^\nu\|$ is $o(\|x^\nu - x^*\|_H)$. Specifically, we can extract the following theorem from Iusem (1991).

**Theorem 6.** *Iusem (1991) Let $x^*$ is the optimal solution for problem 6.1 and $\tilde{x}^n$ is the sequence produced by using the same sequence of hyperplanes but with $\tilde{f}$ instead of $f$. Given a sequence $x^n$ produced by Bregman projections, such that $x^n \to x^*$, and for large enough $\nu$ we satisfy assumptions (1), (2), and (3), then $\|x^\nu - \tilde{x}^\nu\|$ is $o(\|x^\nu - x^*\|_H)$*

Using this we can get the general result as follows

$$
\begin{aligned}
\|x^{\nu+1} - x^*\|_H &\leq \|x^{\nu+1} - \tilde{x}^{\nu+1}\|_H + \|\tilde{x}^{\nu+1} - x^*\|_H \\
&\leq \|x^{\nu+1} - \tilde{x}^{\nu+1}\|_H + \rho\|\tilde{x}^\nu - x^*\|_H && \text{[Quadratic case convergence]} \\
&\leq \|x^{\nu+1} - \tilde{x}^{\nu+1}\|_H + \rho\|\tilde{x}^\nu - x^\nu\|_H + \rho\|x^\nu - x^*\|_H
\end{aligned}
$$

Then diving by $\|x^{\nu+1} - x^*\|_H$, and using Theorem 6 to take the limit, we get that there exists $\rho \in (0, 1)$ such that

$$
\lim_{\nu \to \infty} \frac{\|x^* - x^{\nu+1}\|_H}{\|x^* - x^\nu\|_H} \leq \rho \tag{8.3}
$$

As with the quadratic case, we see that is an oracle satisfies property 2, then 8.3 holds with probability 1. Thus, we have proved Theorem 3 in its complete generality.

### 8.4 PROOF OF THEOREM 4

In this section we prove Theorem 4 in essentially the same manner as we did for Theorem 3 and so we outline only what changes are necessary.

**Theorem 4.** *If $f \in \mathcal{B}(S)$, the hyperplanes $H_i$ are strongly zone consistent with respect to $f$, and $\exists x^0 \in S$ such that $\nabla f(x^0) = 0$, then with probability 1 any sequence $x^n$ produced by the algorithm converges to the optimal solution of problem 6.1. Furthermore, if $x^*$ is the optimal solution, $f$ is twice differentiable at $x^*$, and the Hessian $H := Hf(x^*)$ is positive semidefinite, then there exists $\rho \in (0, 1)$ such that with probability 1,*

$$
\liminf_{\nu \to \infty} \frac{\|x^* - x^{\nu+1}\|_H}{\|x^* - x^\nu\|_H} \leq \rho.
$$

To prove Theorem 4, we need to analyze only what goes wrong if the algorithm "forgets" all of the old constraints. First, consider the proof in the case that we converge to the optimal solution. Then, steps 1,2, and 3 are completely unaffected by forgetting old constraints. The only step that is affected is step 4. In a previous proof, we argued that if for some inactive constraint $a_i$, $z_i$ is non-zero infinitely often, then we projected onto this constraint infinitely often. In our present setting, we cannot conclude this directly as $z_i^\nu > 0$ does not imply that we remember $a_i$ on the $\nu$th iteration. However, due to property 2, we know that $\mathcal{Q}$ returns $a_i$ with probability at least $\tau$. Thus, again using the Borel Cantelli Lemmas, we see that we have $a_i$ infinitely often and this iteration converges to the optimal.

To prove the second part of the theorem, we recall from Theorem 6 that we only need to analyze the case when $f$ is a quadratic function. Indeed, the only place where we used the fact that we remembered old constraints was in the proof of Theorem 5 in which we needed to remember old constraints to guarantee that during the $\nu$th iteration we project onto the constraint $a_i$ that is furthest from $x^\nu$ among those constraints for which $z_i^{\nu+1} > 0$. We cannot guarantee that this happens always but we can guarantee that it happens infinitely often.

Therefore, the conclusion of Theorem 5 holds infinitely often instead of for the tail of the sequence and we replace the limit with a limit infimum to obtain the desired result.

## 9 Application Details

All code, data, and outputs from the experiments can be found at `https://www.dropbox.com/sh/lq5nnhi4je2lh89/AABUUW7k5z3lXTSm8x1hhN1Da?dl=0`. All code was written in Julia 1.1.0 and run on Google Cloud Compute instances.

### 9.1 Metric Nearness

**Convergence Criterion.**

One variant of the metric nearness problem is the decrease only variant, in which we are not allowed to increase the distances and must only decrease them. This problem can solved in $O(n^3)$ time by calculating the all pairs shortest path metric Gilbert & Jain (2017). Given $x^n$ as input, let $\hat{x}^n$ be the optimal decrease only metric. We ran these experiments until $\|\hat{x}^n - x^n\|_2 \leq 1$.

**Implementations.**

We implemented the algorithm from Brickell et al. (2008). We made a small modification that improves the running time. In Brickell et al. (2008), it is recommended that we store the dual variable $z$ as a sparse vector. However, as we do not want the overhead of handling sparse vectors, we store $z$ as a dense vector.

Algorithm 1 was implemented with two modifications. As we can see from algorithm **??**, when the oracle finds violated constraints, it looks at each edge in $G$ and then decides whether there is a violated inequality with that edge. It is cleaner in theory to find all such violated constraints at once and then do the project and forget steps. It is, however, much more efficient in practice to do the project and forget steps for a single constraint as we find it. This approach also helps cut down on memory usage.

The second modification is that once our oracle returns a list of constraints (note we have already projected onto these once), we project onto our whole list of constraints again. Thus, for the constraints returned by the oracle, we project onto these constraints twice per iteration. Note this does not affect any of the convergence results for the algorithm. The pseudocode for this modification can be seen in Algorithm 5.

---

**Algorithm 5** Pseudocode for the implementation for Metric Nearness.

1:  $L^0 = \emptyset$, $z^0 = 0$. Initialize $x^0$ so that $\nabla f(x^0) = 0$.
2:  **while** Not Converged **do**
3:      Let $d(i, j)$ be the weight of shortest path between nodes $i$ and $j$ or $\infty$ if none exists.
4:      $L = \emptyset$
5:      **for** Edge $e = i(, j) \in E$ **do**
6:          **if** $w(i, j) > d(i, j)$ **then**
7:              Let $P$ be the shortest path between $i$ and $j$.
8:              Let $C = P \cup \{(i, j)\}$.
9:              Project onto $C$ and update $x, z$.
10:              **if** $z_C ! = 0$ **then**
11:                  Add $C$ to $L$.
12:      $\tilde{L}^{\nu+1} = L^\nu \cup L$
13:      $x^{\nu+1}, z^{\nu+1} = \text{Project}(x^\nu, z^\nu, \tilde{L}^{\nu+1})$
14:      $L^{\nu+1} = \text{Forget}(\tilde{L}^{\nu+1})$
    **return** $x$

---

**Additional Test Case.**

We also tested our algorithm on an additional type of random weighted complete graph. Let $u_{ij}$ be sampled from the uniform distribution on $[0, 1]$ and $v_{ij}$ from a standard normal, then the weight for an edge $e = ij$ is given by

$$w_{ij} = \lceil 1000 \cdot u_{ij} \cdot v_{ij}^2 \rceil$$

In this case, we got the following running times.

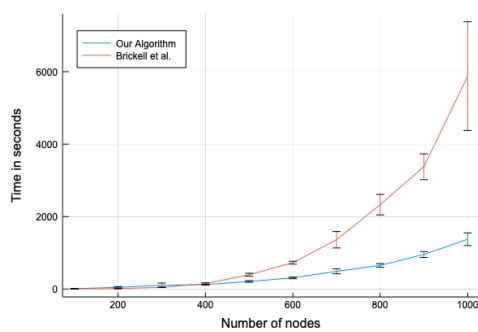

Figure 3: The red line is the mean running time for the algorithm from Brickell et al. (2008). The blue line is the running mean time for our algorithm. All computations were done on a machine with 4 physical cores, each with 13 GB of RAM.

## 9.2 Correlation Clustering

**Transforming the LP.** The formulation of the LP that we solve is as follows:

$$\begin{array}{ll}
\text{minimize} & \tilde{w}^T f + \frac{1}{\gamma} f^T \cdot W \cdot f \\
\text{subject to} & x \in \text{MET}(G) \\
& f_{ij} = |x_{ij} - d_{ij}|, \quad (i,j) \in E.
\end{array} \tag{9.1}$$

The transformation 2.2 into 9.1 has two parts. The first part is the transformation done in Veldt et al. (2019) to obtain the formulation presented in 4.1. To do this transformation we define $\tilde{w}(e) = |w^+(e) - w^-(e)|$. Then $W$ is a diagonal matrix whose entries are given by $\tilde{w}$. Finally, we define $d$ as follows

$$d_{ij} = \begin{cases} 1 & w^{-1}(e) > w^+(e) \\ 0 & \text{otherwise} \end{cases}$$

The second step of the transformation, is the relaxation from $x \in \text{MET}_n$ to $x \in \text{MET}(G)$. The proof for the second is now presented.

**Proposition 2.** *Let $f(x)$ be a function whose values only depends on the values $x_{ij}$ for $e = (i,j) \in G$ and consider the following constrained optimization problem.*

$$\begin{array}{ll}
\text{minimize} & f(x) \\
\text{subject to} & x \in \text{MET}(K_n)
\end{array} \tag{9.2}$$

*Let $\pi$ be the projection from $\text{MET}_n$ to $\text{MET}(G)$ and let $\tilde{f}(\pi(x)) = f(x)$. Then for any optimal solution $x^*$ to the following problem*

$$\begin{array}{ll}
\text{minimize} & \tilde{f}(x) \\
\text{subject to} & x \in \text{MET}(G)
\end{array} \tag{4.2}$$

*we have that for all $\hat{x} \in \pi^{-1}(x^*)$, $\hat{x}$ is an optimal solution to 9.2.*

*Proof.* Here, we see that if $\tilde{x}$ is the minimizer of 4.2 and $x^*$ the minimizer of 9.2 then

$$f(x^*) = \tilde{f}(\pi(x^*)) \geq \tilde{f}(\tilde{x}) = f(\pi^{-1}(\tilde{x}))$$

$\square$

**Calculating the Approximation Ratio.**

Let $\hat{x}$ be the optimal solution to 4.1, then if we let $R = \dfrac{\hat{x}^T W \hat{x}}{2\gamma \tilde{w}^T \hat{x}}$, by Veldt et al. (2019), we have that $\hat{x}$ is an $\dfrac{1+\gamma}{1+R}$ approximation to the optimal solution of 2.2. This is the formula we used to calculate the approximation ratios reported in Tables 1 and 2. For our experiments we used $\gamma = 1$.

**Convergence Criterion.**

We ran the experiment until the maximum violation of a metric inequality was at most 0.01. However, the two algorithms, Ruggles et al. (2019) and ours, have different metric constraints. Specifically Ruggles et al. (2019) only uses all constraints that come from 3 cycles, whereas we use all cycle constraints. Theoretically both sets of constraints define the same polytope, but practically there is a difference. Thus, in practice our algorithm was run to a slightly greater level of convergence than the one from Ruggles et al. (2019).

**Implementations.**

For the case when $G = K_n$, in addition to the modifications that were done for metric nearness experiment, we made two more modifications. First, we did the PROJECT and FORGET step one additional time per iteration. Second, we parallelized the oracle by running Dykstra's algorithm in parallel. The pseudocode for this version of algorithm 1 can be seen in Algorithm 6.

---

**Algorithm 6** Pseudocode for the implementation for CC for the dense case.

---

1: $L^0 = \emptyset$, $z^0 = 0$. Initialize $x^0$ so that $\nabla f(x^0) = 0$.
2: $L_a$ is the list of additional constraints. $z_a^0 = 0$ (dual for additional constraints)
3: **while** Not Converged **do**
4:     Let $d(i,j)$ be the weight of shortest path between nodes $i$ and $j$ or $\infty$ if none exists. This is found using a parallel algorithm.
5:     $L = \emptyset$
6:     **for** Edge $e = i(,j) \in E$ **do**
7:         **if** $w(i,j) > d(i,j)$ **then**
8:             Let $C = P \cup \{(i,j)\}$. Where $P$ be the shortest path between $i$ and $j$.
9:             Project onto $C$ and update $x, z$.
10:             **if** $z_C ! = 0$ **then** Add $C$ to $L$.
11:     $L^\nu \leftarrow L^\nu \cup L$
12:     **for** $i = 1, 2$ **do**
13:         $x^\nu, z^\nu \leftarrow \text{Project}(x^\nu, z^\nu, L^\nu)$
14:         $L^\nu \leftarrow \text{Forget}(L^\nu)$
15:     $x^\nu, z_a^\nu \leftarrow \text{Project}(x^\nu, z_a^\nu, L_a)$
16:     $x^{\nu+1} = x^\nu, L^{\nu+1} = L^\nu, z^{\nu+1} = z^\nu, z_a^{\nu+1} = z_a^\nu,$
    **return** $x$

---

For the sparse version, we made only two modifications: we used the parallel version of the oracle, and during each iteration, we did the project and forget step 75 times.

Note that for both experiments, the additional constraints that were introduced due to the transformation were all projected onto once per iteration and never forgotten. The pseudocode for this version can be seen in Algorithm 7.

We used the implementation provided by the authors of Veldt et al. (2019) to run the experiments for their algorithm.

### 9.3 INFORMATION THEORETIC METRIC LEARNING

The hyper-parameters were set as follows: $\gamma = 1, u = 1, l = 10$. The pseudocode for our algorithm can be seen in Algorithm 8. The classification was done using the $k$ nearest neighbor classifier.

---

**Algorithm 7** Pseudocode for the implementation for CC for the sparse case.

---

1: $L^0 = \emptyset$, $z^0 = 0$. Initialize $x^0$ so that $\nabla f(x^0) = 0$.
2: $L_a$ is the list of additional constraints. $z_a^0 = 0$ (dual for additional constraints)
3: **while** Not Converged **do**
4:      Let $d(i, j)$ be the weight of shortest path between nodes $i$ and $j$ or $\infty$ if none exists. This is found using a parallel algorithm.
5:      $L = \emptyset$
6:      **for** Edge $e = i(, j) \in E$ **do**
7:          **if** $w(i, j) > d(i, j)$ **then**
8:              Let $C = P \cup \{(i, j)\}$. Where $P$ be the shortest path between $i$ and $j$.
9:              Add $C$ to $L$.
10:      $L^\nu \leftarrow L^\nu \cup L$
11:      **for** $i = 1, \ldots, 75$ **do**
12:          $x^\nu, z^\nu \leftarrow \text{Project}(x^\nu, z^\nu, L^\nu)$
13:          $L^\nu \leftarrow \text{Forget}(L^\nu)$
14:      $x^\nu, z_a^\nu \leftarrow \text{Project}(x^\nu, z_a^\nu, L_a)$
15:      $x^{\nu+1} = x^\nu, L^{\nu+1} = L^\nu, z^{\nu+1} = z^\nu, z_a^{\nu+1} = z_a^\nu,$
     **return** $x$

---

**Algorithm 8** Pseudocode for the Project and Forget algorithm for ITML.

---

1: **function** PFITML$(X, C, \gamma, u, l, S, D)$
2:      $\lambda^0 = 0$, $\Xi_{ij} = u$ for $(i, j) \in S$ and $\Xi_{ij} = l$ for $(i, j) \in D$. Initialize $C = I$.
3:      **while** Not Converged **do**
4:          Randomly sample $(i, j)$ from $S$
5:          Do projection for this constraint
6:          Randomly sample $(i, j)$ from $D$
7:          Do projection for this constraint
         **return** $C$
8: **function** PROJECTION$(X, i, j, S, D, u, l, \Xi, \lambda, C)$
9:      $p = dist_C(X_i, X_j)$
10:      $\delta = 1$ if $(i, j) \in S$ and $\delta = -1$ if $(i, j) \in D$
11:      $\alpha = min\left(\lambda_{ij}, \frac{\delta}{2}\left(\frac{1}{p} - \frac{\gamma}{\Xi_{ij}}\right)\right)$
12:      $\beta = \frac{\delta\alpha}{1 - \delta\alpha p}$
13:      $\Xi_{ij} = \gamma\frac{\Xi_{ij}}{\gamma + \delta\alpha\Xi_{ij}}$
14:      $\lambda_{ij} = \lambda_{ij} - \alpha$
15:      $C = C + \beta C(x_i - x_j)(x_i - x_jj)^T C^T$

---

