# OpenReview forum: "Project and Forget: Solving Large Scale Metric Constrained Problems"
_ICLR.cc/2020/Conference — Reject_

### Official Review · AnonReviewer3 · 2019-10-21
**Official Blind Review #3**

**Rating:** 6

**Review:**

This paper proposes a new method for solving the metric constrained problem based on projections on cutting planes. Its main contribution comes from the "forgetting" part, where unnecessary constraints (that are inactive) are removed in order to keep the number of constraints manageable.

Pros:

The methods seem practically useful as verified in the experiments.

Cons:

Most importantly, the paper is out of format and there exist some critical typos that need to be fixed.
- The margin of the paper is wider than the official ICLR format. It needs to be reformatted and verified to be under 10 pages limit.
- There seem to be multiple Latex bugs on referring the section numbers, e.g., "see appendix refsec:genealProblem" at bottom of page 5.

There is no theoretical guarantee on its improvement over existing methods, i.e., the forgotten constraints can reappear during optimization for multiple numbers of times. However, I think this point is not crucial given the empirical usefulness of the algorithm.

Minor questions:
- To my knowledge, cutting plane methods for the integer programming method (including Gurobi) already use an instance "project and forget" method, i.e., iteratively solving linear programs and then adding & removing cutting planes. See [1] for an example. Could the authors discuss the relationship between the two methods and highlight the relative difference & contribution?

[1] The cutting plane method is polynomial for perfect matchings, Chandrasekaran et al., 2012

=========

I have checked that the authors have re-formatted the paper into a correct form. I raise my score since I think the paper is interesting and provides a practically useful algorithm.

**Experience Assessment:**

I do not know much about this area.

**Review Assessment: Checking Correctness Of Derivations And Theory:**

I did not assess the derivations or theory.

**Review Assessment: Checking Correctness Of Experiments:**

I assessed the sensibility of the experiments.

**Review Assessment: Thoroughness In Paper Reading:**

I made a quick assessment of this paper.

---

> ### Author Response · Authors · 2019-11-06
> **Fixed Formatting issues**
>
> Thank you for pointing out the formatting issue. This was an oversight on our part. One of the packages we imported unintentionally changed the margin sizes. The formatting issue has now been fixed and the paper stays within the page limit.

---

> > ### Comment · AnonReviewer3 · 2019-11-06
> > **Thank you for the fast response**
> >
> > I have checked that the paper is within the page limit. Maybe the authors could benefit from making the paper more abstract, i.e., limiting it to 8 pages. Nevertheless, I will raise my score since I find the method to be interesting and practically useful (potentially with broader application).

---

> ### Author Response · Authors · 2019-11-08
> **Relation to Cutting Planes**
>
> The question about cutting planes is a great question. As far as this author understands it, the cutting plane method in Gurobi is for mixed integer programming (MIP) and uses Gromov cuts.
>
> The difficulty of MIPs comes from the integrality constraints and not because they have a large number of constraints. Hence using Gromov cuts lets us reduce the choices for the integral constraints until we can add the constraint $x=c$. Thus, reducing it to a linear program.
>
> However, in general, the cutting plane method highly depends on the choice of cuts.  See [2,3] for in-depth discussions. For other problems that are not MIPs, we have some success cases such as [1], but we have not had success in all problems.
>
> One of the key differences between the standard cutting plane method and Project and Forget is that in the standard cutting plane method, every time we introduce new constraints, we solve the whole optimization problem again. In the case of Project and Forget, we do a round of projections.
>
> Re-solving the LP again has potential issues. In the case of metric constrained problems, we have a large number of constraints. Thus, if we added a large portion of these constraints, we still cannot solve the LP using standard techniques.  We could add in the constraints slowly so that this is not an issue. However, then the intermediate solutions do not necessarily tell us anything about the final solution, and it is unclear whether progress is being made, and we may need too many rounds.
>
> The Project and Forget method addresses this issue. If we added into many inactive constraints, then we only need to do one round of projections. This is less computationally expensive than solving a whole LP. Thus, we get to the forget step much faster. Thus in practice, we add a large number of constraints initially, as seen in Figure 1. However, we forget the inactive constraints quickly, and the projections done onto the active constraints constitute some progress towards finding the final solution. (We may have projected onto inactive constraints, but experimentally, we tend to undo this relatively quickly.)
>
> Finally, we also have the generic version of our algorithm presented in the appendix. For the general version of the algorithm, we have a subroutine that we dub the oracle. Here the oracle is just the cutting plane selection method. Here we show that under some weak assumptions (property 1) that our algorithm has a linear rate of convergence. Additionally, we show that if we randomly sample constraints, then with probability 1, we have a linear rate of convergence.
>
> [1]Karthekeyan Chandrasekaran, László A. Végh, and Santosh Vempala. The cutting plane method is polynomial for perfect matchings. In 2012 IEEE 53rd Annual Symposium on Foundations of Computer Science
> [2]Santanu S. Dey and Marco Molinaro. Theoretical challenges towards cutting-plane selection. Math. Program.
> [3]Laurent Poirrier and James Yu. On the depth of cutting planes. arXiv e-prints.

---

### Official Review · AnonReviewer1 · 2019-10-24
**Official Blind Review #775**

**Rating:** 3

**Review:**

The paper presents an algorithm for optimizing an function f under the constraints that the square matrix variable x represents "metric". In this context, this means that we have also observed a graph G with n vertices, and x is of size n by n, x(i, j) < x(i, e) + x(e, j) if i ~ e and j ~ e are adjacent: this is a generalized for of triangle inequality.
Authors argue that the constraint "x is a metric" translate into exponentially many linear constraints, which results in to a hard to solve problem
The algorithm they propose to tackle this (Algorithm 1) has two subroutines that are shown in Algorithm 2 (Forget and Project). The Project subroutine itself is a projection onto a convex set according to a Bregman divergence, which is not trivial. In this paper I understand that authors only consider metrics of type x' L x where L = C'C >0 is psd

Authors claim that the sequence created by their algorithm asymptotically converges to the global optimum, and show numerical superiority to baselines.

Major remarks:

My general feeling is that the paper overstates its results. The paper has some good contribution, which could be better emphasized.

The algorithm stacks multiple subroutines which are not necessarily very light. I am skeptical about the numerical efficiency of such algorithms.

Theoretical results are stated asymptotically while interpreted in the text as finite steps results: page 5, after Corollary 1., read "The algorithm spends the first few iterations ..." in this case, a theoretical result should support the claim

The algorithm starts at a stationary point of f. This itself can be nontrivial. Can authors discuss this?

Minor remarks:

metric and distance to me mean the same, hence the first sentence of the intro doesn't read easily..

what is \cal A line 5 of Algorithm 1? It seems to be a "list of hyperplanes" according to the previous text, but it is unclear to me how to build it algorithmically

The notation L is confusing in Algo 1 MetricViolation: wasn't L the matrix defining the metric?

A few typos: l. 12 Algo 1, e = (i, j), 3.2 "global optimum [remove solution]."



**Experience Assessment:**

I have read many papers in this area.

**Review Assessment: Checking Correctness Of Derivations And Theory:**

I did not assess the derivations or theory.

**Review Assessment: Checking Correctness Of Experiments:**

I did not assess the experiments.

**Review Assessment: Thoroughness In Paper Reading:**

I read the paper at least twice and used my best judgement in assessing the paper.

---

> ### Author Response · Authors · 2019-11-08
> **Minor remarks:**
>
> 1) This list should be part of the input. The pseudocode has been updated to reflect this.
> 2) L was the matrix. We will change that letter in the metric learning definition.
> 3) Typos have been fixed.

---

> ### Author Response · Authors · 2019-11-08
> **Theoretical Results**
>
> "Theoretical results are stated asymptotically while interpreted in the text as finite steps results . . . theoretical result should support the claim."
>
> If you look at the statement of proposition 1, it says that that for all inactive constraints $a_i$, we have that $z_i^\nu = 0$ for the tail. Since this is true for the whole tail, this implies that there is a finite time N  after which $z_i = 0$. We have finitely many constraints, so there is a finite time, after which we the z's are 0 for all inactive constraints. This means that we have forgotten all of the inactive constraints. Thus, the first few iterations are these finite iterations. From a practical standpoint (as seen in Figure 1, this happens very quickly). For the sake of clarity, we have rewritten the proposition, so it says this instead.

---

> ### Author Response · Authors · 2019-11-08
> **Numerical efficiency**
>
> The first most crucial evidence for the numerical efficiency of the algorithm is our experimental results. As both reviewers 2 and 3  highlight, our experimental results point to the practical usefulness of our algorithm.
>
> Next, we would like to address the specific concerns that were brought up.
>
> 1) "The Project subroutine itself is a projection onto a convex set according to a Bregman divergence, which is not trivial."
>
> In terms of computing Bregman projections, we don't need to project onto general convex sets. We only need to project onto hyperplanes. This makes the problem of computing the projection much easier. Note in the project subroutine we don't project onto the intersection of all of the halfspaces. We project onto each hyperplane separately iteratively.
>
> Additionally, for standard objective functions such as a quadratic, we have an analytic formula for the projection. Thus can be done in constant time. Additionally, in the case of metric constraints, the constraints tend to be sparse (small cycles). Thus, even the numerical approximation algorithm presented in [2] is extremely fast.
>
> 2) "The algorithm stacks multiple subroutines which are not necessarily very light. I am skeptical about the numerical efficiency of such algorithms."
>
> We only have two subroutines that could be nontrivial: the metric violations and the project routine, the forget routine is relatively simple. We have already addressed the project routine, so we will now look at the routine the metric violations routine.
>
> Any algorithm to solve such problems would have to deal with the fact that we have $n^3$ constraints. From previous work [1], we know that standard methods run out of memory due to the large number of constraints. Thus, we need to use cyclic iterative methods that need to cycle through all $n^3$ constraints. In terms of $n^3$ running time, the Floyd Warshall algorithm is one of the simplest and fastest $n^3$ algorithms. In particular, it is faster than having to do all $n^3$ projections. We can see this due to the metric nearness experiment.
> Also, the Floyd Warshall algorithm is one of the "stupidly parallelizable" algorithms. Hence we can parallelize this computation.
>
> In the case when the graph is sparse, the metric violations can be solved faster than $n^3$. Hence allowing us to solve correlation clustering on instances 10x bigger than previously done.
>
> 3)  "The algorithm starts at a stationary point of f. This itself can be nontrivial. Can authors discuss this?"
>
> In many cases, we can theoretically calculate what the stationary point is. If f(x) is a quadratic = $x^TQx + c^Tx + a$ where Q is positive definite then the stationary point is given by $-Q^{-1}c$. If f(x) is entropy $-\sum_i x_i log(x_i)$, then the stationary point is given by x is the vector where every coordinate is $1/e$.
>
> In general, we are assuming that our function is a Bregman function, which means that f(x) is strictly convex. Thus, if we cannot theoretically compute this, we can numerically approximate this via gradient descent or Newton's method. Such a computation only needs to be done once.
>
> [1] Nate Veldt, David Gleich, Anthony Wirth, and James Saunderson. Metric-constrained optimization for graph clustering algorithms. SIAM Journal on Mathematics of Data Science
> [2] Inderjit S. Dhillon and Joel A. Tropp. Matrix nearness problems with Bregman divergences. SIAM J. Matrix Anal. Appl.

---

> ### Author Response · Authors · 2019-11-08
> **Misunderstanding of our work**
>
> Thank you for taking the time to read our work. However, we would be extremely grateful if you took a look at our experiments as the experiments are the main strength of our paper.
>
> One of the main ideas of our work is that we can solve optimization problems over the whole metric polytope. Thus, we do not have to restrict ourselves to metrics of a specific type. In particular, the metrics learned in metric nearness and correlation clustering are not of the type x’ L x. We only restrict ourselves to that for the metric learning problem because of the formulation of the information-theoretic metric learning objective.
>
> One of the avenues of future research we hope to open up is precisely taking these formulations where we restrict to metrics of the type x’L x and generalizing it to learn a general metric using Project and Forget.

---

### Official Review · AnonReviewer2 · 2019-10-30
**Official Blind Review #2**

**Rating:** 6

**Review:**

The paper considers the problem of optimizing convex functions under metric constraints. The main challenge is that expressing all metric constraints on n points requiries O(n^3) constraints. The paper proposes a “project and forget” approach which is essentially is based on cyclic Bregman projections but with a twist that some of the constraints are forgotten. The proof of convergence of this method is given, but no explicit bound on the number of iterations. While the general method doesn’t appear to be particularly novel, I found it quite impressive that the authors were able to solve 10x larger instances of weighted correlation clustering than the previous work. While from a theoretical perspective this work is hardly very exciting, the practical results are rather interesting. Other applications to the metric nearness and metric learning problems are also given.


Comments:
-- The paper is full of typos and needs to be proofread by a native English speaker.


**Experience Assessment:**

I have published one or two papers in this area.

**Review Assessment: Checking Correctness Of Derivations And Theory:**

I assessed the sensibility of the derivations and theory.

**Review Assessment: Checking Correctness Of Experiments:**

I assessed the sensibility of the experiments.

**Review Assessment: Thoroughness In Paper Reading:**

I read the paper at least twice and used my best judgement in assessing the paper.

---

> ### Author Response · Authors · 2019-11-08
> **Theoretical Contributions**
>
> Thank you for taking the time and reading our work. We agree that the main strength of our paper lies in the experimental results that we have obtained. However, that is not to say that the theoretical results are unimportant.
>
> The Bregman method has existed for a long time, and lots of research work has been done on the method. However, until now, all Bregman algorithms were constrained at cyclically (or almost cyclically) looking at the constraints. In fact, the need to cyclically look at the constraints to show that the algorithm converges to the optimal solution is an aspect that is highlighted in previous work. See [1,2,3] for examples.
>
> Many applications that used these methods found other ways around needing to see all the constraints. This was done either by restricting the number of constraints and solving a heuristic problem, by solving smaller sized problems, or by trying to parallelize the projections. See [4,5,6] for examples of each. Thus, to prove the convergence result while incorporating the ability to add new constraints and to forget old constraints is vital. Without having the convergence, the increased speed obtained from doing these steps could be useless.
>
> [1] Yair Censor and Simeon Reich. The Dykstra Algorithm with Bregman Projections. Communications in Applied Analysis.
> [2] Heinz H. Bauschke and Adrian S. Lewis. Dykstra’s algorithm with Bregman projections: a convergence proof. Optimization.
> [3] Yair Censor and Stavros Zenios. Parallel optimization: Theory, algorithms, and applications.
> [4] Jason V. Davis, Brian Kulis, Prateek Jain, Suvrit Sra, and Inderjit S. Dhillon. Information-theoretic metric learning. In Proceedings of the 24th International Conference on Machine Learning.
> [5] Justin Brickell, Inderjit S. Dhillon, Suvrit Sra, and Joel A. Tropp. The metric nearness problem. SIAM J. Matrix Anal.
> [6] Cameron Ruggles, Nate Veldt, and David F. Gleich. A Parallel Projection Method for Metric Constrained Optimization. arXiv e-prints.

---

### Author Response · Authors · 2022-11-29
**Paper Accepted at JMLR**

This paper has now been accepted at JMLR. Hence I adding a pointer here to the published version. https://jmlr.org/papers/v23/20-1424.html

---

### Decision · Program_Chairs · 2019-12-19

**Decision:**

Reject

**Comment:**

Quoting from Reviewer2: "The paper considers the problem of optimizing convex functions under metric constraints. The main challenge is that expressing all metric constraints on n points requiries O(n^3) constraints. The paper proposes a “project and forget” approach which is essentially is based on cyclic Bregman projections but with a twist that some of the constraints are forgotten."  The reviewers were split on this submission, with two arguing for weak acceptance and one arguing for rejection.  Purely based on scores, this paper is borderline.  It was pointed out by multiple reviewers that the method is not very novel.  In particular it effectively works as an active set method.  It appears to be very effective in this setting, but the basic algorithm does not differ in structure from any active set method, for which removal of inactive constraints is considered standard (see even the wikipedia page on active set methods).

---

> ### Author Response · Authors · 2019-12-23
> **Active Set**
>
> Yes, the method is a type of active set method. The novelty comes being able to combine the idea behind active sets and an iterative method such Bregman's cyclic projections method. The result of this is that when our initial guess for active constraints is really large, (e.g.  As seen in Figure 1, for Ca-HepTh we have at at least 10^{7.5} violated constraints in the beginning) then Project and Forget is still viable whereas previous active set methods are not.
>
> Additionally, this setting of metric constrained problems is an important setting. Hence the applicability of our method here is important, as other previously known methods have proved to be not as effective in this setting.